# Learning Generative Vision Transformer with Energy-Based Latent Space for Saliency Prediction

Jing Zhang[1], Jianwen Xie[1], Nick Barnes[2], Ping Li[1]

[1] Cognitive Computing Lab, Baidu Research
[2] The Australian National University

{zjnwpu, jianwen.kenny, pingli98}@gmail.com, nick.barnes@anu.edu.au

## Abstract

Vision transformer networks have shown superiority in many computer vision tasks. In this paper, we take a step further by proposing a novel generative vision transformer with latent variables following an informative energy-based prior for salient object detection. Both the vision transformer network and the energy-based prior model are jointly trained via Markov chain Monte Carlo-based maximum likelihood estimation, in which the sampling from the intractable posterior and prior distributions of the latent variables are performed by Langevin dynamics. Further, with the generative vision transformer, we can easily obtain a pixel-wise uncertainty map from an image, which indicates the model confidence in predicting saliency from the image. Different from the existing generative models which define the prior distribution of the latent variables as a simple isotropic Gaussian distribution, our model uses an energy-based informative prior which can be more expressive to capture the latent space of the data. We apply the proposed framework to both RGB and RGB-D salient object detection tasks. Extensive experimental results show that our framework can achieve not only accurate saliency predictions but also meaningful uncertainty maps that are consistent with the human perception.

## 1 Introduction

In the field of computer vision, salient object detection [64, 65, 16, 17, 5, 89] (SOD) or visual saliency prediction, which aims at highlighting objects more attentive than the surrounding areas in images, has achieved significant performance improvement with the deep convolutional neural network revolution. Given a set of training images along with their saliency annotations, the conventional SOD models seek to learn a deterministic one-to-one mapping from image domain to saliency domain.

Two main issues exist in the above conventional deep saliency prediction framework: (i) the convolution operation based on sliding window makes the deep saliency prediction model less effective in modeling the global contrast of the image, which is essential for salient object detection [7]; (ii) the one-to-one deterministic mapping mechanism makes the current framework not only impossible to represent the pixel-wise uncertainty in predicting salient objects, but also hard to handle incomplete data in a weakly supervised scenario [89]. Besides, given an image, the saliency output of a human is subjective, therefore, a stochastic generative model is more natural than a deterministic model for representing saliency prediction. Although [85] introduces a conditional variational autoencoder (CVAE) [56] for RGB-D salient object detection, the potential posterior collapse problem [23] makes the stochastic predictions less effective in generating meaningful uncertainty estimation.

Motivated by the above two issues, we propose a novel framework, the generative vision transformer, for salient object detection, where a vision transformer structure [40] is used as a backbone and latent variables are introduced in designing our generative framework. On the one hand, transformers [60] have proven to be very effective in long-range dependency modeling, and are capable of modeling

various scopes of object context information with the multi-head self-attention module. With such an architecture, we can achieve global context modeling for effective salient object detection. On the other hand, the latent variables account for randomness and uncertainty in modeling the mapping from image domain to saliency domain, and also enable the model to produce stochastic saliency predictions for uncertainty estimation. Therefore, the proposed model is a latent variable transformer.

Nowadays, there are two types of generative models that have been widely used, namely the variational autoencoder (VAE) [31] and the generative adversarial network (GAN) [20], which correspond to two different generative learning strategies to train latent variable models. To train a top-down latent variable generator, the VAE introduces an extra encoder to approximate the intractable posterior distribution of the latent variables, and trains the generator via a perturbation of maximum likelihood; while the GAN introduces a discriminator to distinguish between generated samples and real data, and trains the generator to fool the discriminator. [22, 69] present the third learning strategy, namely alternating back-propagation (ABP), to train the generator with latent variables being directly sampled from the true posterior distribution by using a gradient-based Markov chain Monte Carlo (MCMC) [38], e.g., Langevin dynamics [43, 66, 12]. All the three generative models define the prior distribution of the latent variables as a simple non-informative isotropic Gaussian distribution, which is less expressive in capturing meaningful latent representation of the data.

In this paper, we investigate generative modeling and learning of the vision transformer. We construct a generative model for salient object detection in the form of a top-down conditional latent variable model. Specifically, we propose a generative vision transformer by adding latent variables into the traditional deterministic vision transformer, and assume the latent variables follow an informative trainable energy-based prior distribution [47, 48]. Following [72], we parameterize the energy function of the energy-based model (EBM) by a deep net. Instead of using variational learning or adversarial learning, we jointly train the parameters of the EBM prior and the transformer network by maximum likelihood estimation (MLE). The MLE algorithm relies on MCMC sampling to evaluate the intractable prior and posterior distributions of the latent variables.

Experimental results on RGB and RGB-D salient object detections [64, 68, 85, 16] show that the generative framework equipped with the EBM prior and the transformer-based non-linear mapping is powerful in representing the conditional distribution of object saliency given an image, leading to more reasonable uncertainty estimation as shown in Figure 1, where stochastic saliency prediction is provided by a learned model and the visualization of the pixel-wise uncertainty is presented.

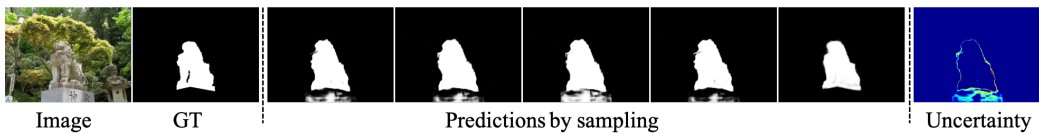

Figure 1: An illustration of the stochastic saliency prediction obtained by the proposed generative vision transformer with an EBM prior, as well as the corresponding pixel-wise uncertainty map.

We summarize our main contributions and novelties as follows: (i) we propose a novel top-down generative vision transformer network with an energy-based prior distribution defined on latent space for salient object detection; (ii) we jointly train the vision transformer network and the energy-based prior model by an MCMC-based maximum likelihood estimation, without relying on any extra assisting network for adversarial learning or variational learning; (iii) we achieve new benchmark results for both RGB and RGB-D salient object detections, and obtain meaningful uncertainty maps that are highly consistent with human perception for saliency prediction.

## 2 Related Work

**Salient object detection:** The main goal of the existing deep fully-supervised RGB image-based salient object detection models [67, 41, 68, 64, 61, 87, 65] is to achieve structure preserving saliency prediction, by either sophisticated feature aggregation [67, 61], auxiliary edge detection [68, 52, 65], or resorting to structure-aware loss functions [41, 64]. With extra depth information, RGB-D salient object detection models [53, 85, 5, 27, 93, 90, 16, 28, 51, 92, 59, 86] mainly focus on effective multi-modal modeling. Our paper solves the same problems, i.e., RGB and RGB-D salient object detection, by developing a new generative transformer-based framework.

**Vision transformers:** The breakthroughs of the Transformer networks [60] in natural language processing (NLP) domain have sparked the interest of the computer vision community in developing vision transformers for different computer vision tasks, such as image classification [10, 40], object detection [4, 63, 6, 40], image segmentation [96, 54, 63, 40], object tracking [80, 81], pose estimation [42, 58], etc. Among them, DPT [54] adopts a U-shape structure and uses ViT [10] as an encoder to perform semantic segmentation and monocular depth estimation. Swin [40] presents a hierarchical transformer with a shifted windowing scheme to achieve an efficient transformer network with high resolution images as input. Different from the above vision transformers that mainly focus on discriminative modeling and learning, our paper emphasizes generative modeling and learning of the vision transformer by involving latent variables and MCMC inference.

**Dense prediction with generative models:** VAEs have been successfully applied to image segmentation [3, 32]. For saliency prediction, [34] adopts a VAE to model the image background, and separates salient objects from the background through reconstruction residuals. [85, 84] design CVAEs to model the subjective nature of saliency. GAN-based methods can be divided into two categories, namely fully-supervised and semi-supervised settings. The former [21, 33] uses the discriminator to distinguish model predictions from ground truths, while the latter [57, 26] uses the GAN to explore the contribution of unlabeled data. [88] uses a cooperative learning framework [71, 74] for generative saliency prediction. [84] trains a single top-down generator in the ABP framework for RGB-D saliency prediction. Our model generalizes [84] by replacing the simple Gaussian prior by a learnable EBM prior and adopting a vision transformer-based generator for salient object prediction.

**Energy-based models:** Recent works have shown strong performance of data space EBMs [72, 44] in modeling high-dimensional complex dependencies, such as images [97, 95, 18, 11, 19], videos [78, 79], 3D shapes [75, 76], and point clouds [73], and also demonstrated the effectiveness of latent space EBMs [47] in improving the model expressivity for text [48], image [47], and trajectory [49] generation. Our paper also learns a latent space EBM as the prior model but builds the EBM on top of a vision transformer generator for image-conditioned saliency map prediction.

## 3 Generative Vision Transformer with Energy-Based Latent Space

### 3.1 Model

We formulate the supervised saliency prediction problem as a conditional generative learning problem. Let $\mathbf{I} \in \mathbb{R}^{h \times w \times 3}$ be an observed RGB image, $s \in \mathbb{R}^{h \times w \times 1}$ be the saliency map, and $z \in \mathbb{R}^{1 \times 1 \times d}$ be the $d$-dimensional vector of latent variables, where $h \times w \gg d$. Consider the following generative model to predict a saliency map $s$ from an image $\mathbf{I}$,

$$s = T_\theta(\mathbf{I}, z) + \epsilon, \quad z \sim p_\alpha(z), \quad \epsilon \sim \mathcal{N}(0, \sigma_\epsilon^2 I), \tag{1}$$

where $T_\theta$ is the non-linear mapping process from $[z, \mathbf{I}]$ to $s$ with parameters $\theta$, $p_\alpha(z)$ is the prior distribution with parameters $\alpha$, and $\epsilon \sim \mathcal{N}(0, \sigma_\epsilon^2 I)$ is the observation residual of saliency with $\sigma_\epsilon$ being given. Due to the stochasticity of the latent variables $z$, given an image $\mathbf{I}$, its saliency map is also stochastic. Such a probabilistic model is in accord with the uncertainty of the image saliency.

Following [47], the prior $p_\alpha(z)$ is not assumed to be a simple isotropic Gaussian distribution as GAN [20], VAE [31, 56] or ABP [22]. Specifically, it is in the form of the energy-based correction or exponential tilting [72] of an isotropic Gaussian reference distribution $p_0(z) = \mathcal{N}(0, \sigma_z^2 I)$, i.e.,

$$p_\alpha(z) = \frac{1}{Z(\alpha)} \exp\left[-U_\alpha(z)\right] p_0(z) \propto \exp\left[-U_\alpha(z) - \frac{1}{2\sigma_z^2}||z||^2\right], \tag{2}$$

where $\mathcal{E}_\alpha(z) = U_\alpha(z) + \frac{1}{2\sigma_z^2}||z||^2$ is the energy function that maps the latent variables $z$ to a scalar, and $U_\alpha(z)$ is parameterized by a multi-layer perceptron (MLP) with trainable parameters $\alpha$. The standard deviation $\sigma_z$ is a hyperparameter. $Z(\alpha) = \int \exp[-U_\alpha(z)] p_0(z) dz$ is the intractable normalizing constant that resolves the requirement for a probability distribution to have a total probability equal to one. $p_\alpha(z)$ is an informative prior distribution in our model and its parameters $\alpha$ need to be estimated along with the non-linear mapping function $T_\theta$ from the training data.

The mapping function $T_\theta$ is parameterized by a vision transformer [40] with self-attention mechanism, which encodes the input image $\mathbf{I}$ and then decodes it along with the vector of latent variables $z$ to the saliency map $s$, thus, $p_\theta(s|\mathbf{I}, z) = \mathcal{N}(T_\theta(\mathbf{I}, z), \sigma_\epsilon^2 I)$. The resulting generative model is a conditional directed graphical model that combines the EBM prior [47] and the vision transformer [40].

## 3.2 Learning

The generative transformer with an energy-based prior, which is presented in Eq. (1), can be trained via maximum likelihood estimation. For notation simplicity, let $\beta = (\theta, \alpha)$. For the training examples $\{(\mathbf{I}_i, s_i), i = 1, ..., n\}$, the observed-data log-likelihood function is defined as

$$L(\beta) = \sum_{i=1}^{n} \log p_\beta(s_i|\mathbf{I}_i) = \sum_{i=1}^{n} \log \left[ \int p_\beta(s_i, z_i|\mathbf{I}_i) dz_i \right] = \sum_{i=1}^{n} \log \left[ \int p_\alpha(z_i) p_\theta(s_i|\mathbf{I}_i, z_i) dz_i \right].$$

Maximizing $L(\beta)$ is equivalent to minimizing the Kullback-Leibler (KL) divergence between the model $p_\beta(s|\mathbf{I})$ and the data distribution $p_{\text{data}}(s|\mathbf{I})$. The gradient of $L(\beta)$ can be computed based on

$$\nabla_\beta \log p_\beta(s|\mathbf{I}) = \mathrm{E}_{p_\beta(z|s,\mathbf{I})} \left[ \nabla_\beta \log p_\beta(s, z|\mathbf{I}) \right] = \mathrm{E}_{p_\beta(z|s,\mathbf{I})}[\nabla_\beta(\log p_\alpha(z) + \log p_\theta(s|\mathbf{I}, z))], \quad (3)$$

where the posterior distribution $p_\beta(z|s, \mathbf{I}) = p_\beta(s, z|\mathbf{I})/p_\beta(s|\mathbf{I}) = p_\alpha(z) p_\theta(s|\mathbf{I}, z)/p_\beta(s|\mathbf{I})$.

The learning gradient in Eq. (3) can be decomposed into two parts, i.e., the gradient for the energy-based model $\alpha$

$$\mathrm{E}_{p_\beta(z|s,\mathbf{I})}[\nabla_\alpha \log p_\alpha(z)] = \mathrm{E}_{p_\alpha(z)}[\nabla_\alpha U_\alpha(z)] - \mathrm{E}_{p_\beta(z|s,\mathbf{I})}[\nabla_\alpha U_\alpha(z)], \quad (4)$$

and the gradient for the transformer $\theta$

$$\mathrm{E}_{p_\beta(z|s,\mathbf{I})}[\nabla_\theta \log p_\theta(s|\mathbf{I}, z)] = \mathrm{E}_{p_\beta(z|s,\mathbf{I})} \left[ \frac{1}{\sigma_\epsilon^2}(s - T_\theta(\mathbf{I}, z))\nabla_\theta T_\theta(\mathbf{I}, z) \right]. \quad (5)$$

$\nabla_\alpha U_\alpha(z)$ in Eq. (4) and $\nabla_\theta T_\theta(\mathbf{I}, z)$ in Eq. (5) can be efficiently computed via back-propagation. Both Eq. (4) and Eq. (5) include intractable expectation terms $\mathrm{E}_p(\cdot)$, which can be approximated by MCMC samples. To be specific, we can use a gradient-based MCMC, e.g., Langevin dynamics, which is initialized with a Gaussian noise distribution $p_0$, to draw samples from the energy-based prior model $p_\alpha(z) \propto \exp\left[-\mathcal{E}_\alpha(z)\right]$ by iterating

$$z_{t+1} = z_t - \delta \nabla_z \mathcal{E}_\alpha(z_t) + \sqrt{2\delta} e_t, \quad z_0 \sim p_0(z), \quad e_t \sim \mathcal{N}(0, I), \quad (6)$$

and draw samples from the posterior distribution $p_\beta(z|s, \mathbf{I})$ by iterating

$$z_{t+1} = z_t - \delta \left[ \nabla_z \mathcal{E}_\alpha(z_t) - \frac{1}{\sigma_\epsilon^2}(s - T_\theta(\mathbf{I}, z_t))\nabla_z T_\theta(\mathbf{I}, z_t) \right] + \sqrt{2\delta} e_t, \quad z_0 \sim p_0(z), \quad e_t \sim \mathcal{N}(0, I).$$
$$(7)$$

$\delta$ is the Langevin step size and can be specified independently in Eq. (6) and Eq. (7). We use $\{z_i^+\}$ and $\{z_i^-\}$ to denote, respectively, the samples from the posterior distribution $p_\beta(z|s, \mathbf{I})$ and the prior distribution $p_\alpha(z)$. The gradients of $\alpha$ and $\theta$ can be computed with $\{(\mathbf{I}_i, s_i)\}$, $\{z_i^+\}$ and $\{z_i^-\}$ by

$$\nabla\alpha = \frac{1}{n} \sum_{i=1}^{n} [\nabla_\alpha U_\alpha(z_i^-)] - \frac{1}{n} \sum_{i=1}^{n} \left[ \nabla_\alpha U_\alpha(z_i^+) \right], \quad (8)$$

$$\nabla\theta = \frac{1}{n} \sum_{i=1}^{n} \left[ \frac{1}{\sigma_\epsilon^2}(s_i - T_\theta(\mathbf{I}_i, z_i^+))\nabla_\theta T_\theta(\mathbf{I}_i, z_i^+) \right], \quad (9)$$

We can update the parameters with $\nabla\alpha$ and $\nabla\theta$ via the Adam optimizer [30]. We present the full learning and sampling algorithm of our model in Algorithm 1.

---

**Algorithm 1** Maximum likelihood learning algorithm for generative vision transformer with energy-based latent space for saliency prediction

---

**Input**: (1) Training images $\{\mathbf{I}_i\}_i^n$ with associated saliency maps $\{s_i\}_i^n$; (2) Number of learning iterations $M$; (3) Numbers of Langevin steps for prior and posterior $\{K^-, K^+\}$; (4) Langevin step sizes for prior and posterior $\{\delta^-, \delta^+\}$; (5) Learning rates for energy-based prior and transformer $\{\xi_\alpha, \xi_\theta\}$; (6) batch size $n'$.
**Output**: Parameters $\theta$ for the transformer and $\alpha$ for the energy-based prior model

1: Initialize $\theta$ and $\alpha$
2: **for** $m \leftarrow 1$ to $M$ **do**
3:     Sample observed image-saliency pairs $\{(\mathbf{I}_i, s_i)\}_i^{n'}$
4:     For each $(\mathbf{I}_i, s_i)$, sample the prior $z_i^- \sim p_{\alpha_m}(z)$ using $K^-$ Langevin steps in Eq. (6) with a step size $\delta^-$.
5:     For each $(\mathbf{I}_i, s_i)$, sample the posterior $z_i^+ \sim p_{\beta_m}(z|s_i, \mathbf{I}_i)$ using $K^+$ Langevin steps in Eq. (7) with a step size $\delta^+$.
6:     Update energy-based prior by Adam with the gradient $\nabla\alpha$ computed in Eq. (8) and a learning rate $\xi_\alpha$.
7:     Update transformer by Adam with the gradient $\nabla\theta$ computed in Eq. (9) and a learning rate $\xi_\theta$.
8: **end for**

---

### 3.3 Network

**Generative vision transformer:** We design the generative vision transformer using the Swin transformer [40] backbone as shown in Figure 2, which takes a three-channel image $\mathbf{I}$ and the latent variables $z$ as input and outputs a one-channel saliency map $T_\theta(\mathbf{I}, z)$. Two main modules are included in our generative vision transformer, including a "Transformer Encoder" module and a "Feature Aggregation" module. The former takes $\mathbf{I}$ as input and produces a set of feature maps $\{f_l\}_{l=1}^5$ of channel sizes 128, 256, 512, 1024 and 1024, respectively, while the latter takes $\{f_l\}_{l=1}^5$ and the vector of latent variables $z$ as input to generate the saliency prediction $s$.

Specifically, we first feed each $f_l$ to a $3 \times 3$ convolutional layer to reduce the channel dimension to 32, and obtain a new set of feature maps $\{f_l'\}_{l=1}^5$ after channel reduction. Then, we replicate the vector $z$ spatially and perform a channel-wise concatenation with $f_5'$, followed by a $3 \times 3$ convolutional layer that seeks to produce a feature map

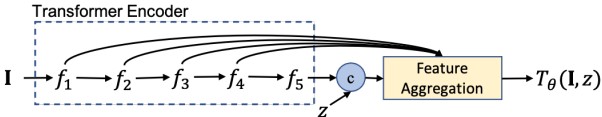

Figure 2: Generative latent variable vision transformer

$F_5$ with same number of channels as that of $f_5'$. Finally, we sequentially concatenate feature maps from high level to low level via feature aggregation, i.e., from $l = 4$ to 1, we compute $F_l = \text{Conv}_{3 \times 3}(\text{M}(\text{Concat}(f_l', F_{l+1}, ..., F_5)))$, where $\text{Conv}_{3 \times 3}(\cdot)$ is a $3 \times 3$ convolutional layer that reduces the channel dimension to 32, $\text{M}(\cdot)$ is the channel attention module [91], and $\text{Concat}(\cdot)$ is the channel-wise concatenation operation. Note that, we upsample the higher level feature map to the same spatial size as that of the lower level feature map before the concatenation operation. We feed $F_1$ to a $3 \times 3$ convolutional layer to obtain the one-channel saliency map $T_\theta(\mathbf{I}, z)$.

**Energy-based prior model:** We design an energy-based model for the latent variables $z$ by parameterizing the function $U_\alpha(z)$ via a multilayer perceptron (MLP), which uses three fully connected layers to map the latent variables $z$ to a scalar. The sizes of the feature maps of different layers of the MLP are $C_e, C_e$ and 1, respectively. We will simply use $C_e$ to represent the size of the EBM prior and set $C_e = 60$ in our experiment. GELU [25] activation is used after each layer except the last one.

### 3.4 Analysis

**Convergence:** Theoretically, when the Adam optimizer of $\beta = (\theta, \alpha)$ in the learning algorithm converges to a local minimum, it solves the following estimating equations

$$\nabla\alpha = 0 \quad \Rightarrow \quad \frac{1}{n}\sum_{i=1}^n \mathrm{E}_{p_\beta(z_i|s_i,\mathbf{I}_i)}[\nabla_\alpha U_\alpha(z_i)] - \mathrm{E}_{p_\alpha(z)}[\nabla_\alpha U_\alpha(z)] = 0, \tag{10}$$

$$\nabla\theta = 0 \quad \Rightarrow \quad \frac{1}{n}\sum_{i=1}^n \mathrm{E}_{p_\beta(z_i|s_i,\mathbf{I}_i)}\left[\frac{1}{\sigma_\epsilon^2}(s_i - T_\theta(\mathbf{I}_i, z_i))\nabla_\theta T_\theta(\mathbf{I}_i, z_i)\right] = 0, \tag{11}$$

which are the maximum likelihood estimating equations. However, in practise, the Langevin dynamics in Eq. (8) and Eq. (9) might not converge to the target distributions due to the use of a small number of Langevin steps (i.e., short-run MCMC), the estimating equations in Eq. (10) and Eq. (11) will correspond to a perturbation of the MLE estimating equation according to [44, 45, 47]. The learning algorithm can be justified as a Robbins-Monro [55] algorithm, whose convergence is theoretically sound. Our model can also be trained with an extra encoder as an amortized inference network [31] for $p_\beta(z|s, \mathbf{I})$ and an extra generator as an amortized sampling network [70, 71, 77] for $p_\alpha(z)$ . In this work, we prefer to keep our training algorithm simple in order to avoid extra efforts for the design of the auxiliary network architectures. We will study the joint training strategy in our future work.

**Accuracy:** Compared with the GAN-based generative framework, our model is a likelihood-based generative framework that will not suffer from mode collapse [2]. In comparison with VAE-based generative framework, whose training is also based on likelihood, our MCMC-based maximum likelihood learning algorithm will not encounter the posterior collapse issue that is caused by amortized inference. On the other hand, the variational inference typically relies on an extra inference network for efficient inference of the latent variables given an image and saliency pair, however, the approximate inference might not be able to take the full place of the posterior distribution in practise. To be specific, we use $q_\phi(z|s, \mathbf{I})$ to denote the tractable approximate inference network with parameters $\phi$. The variational

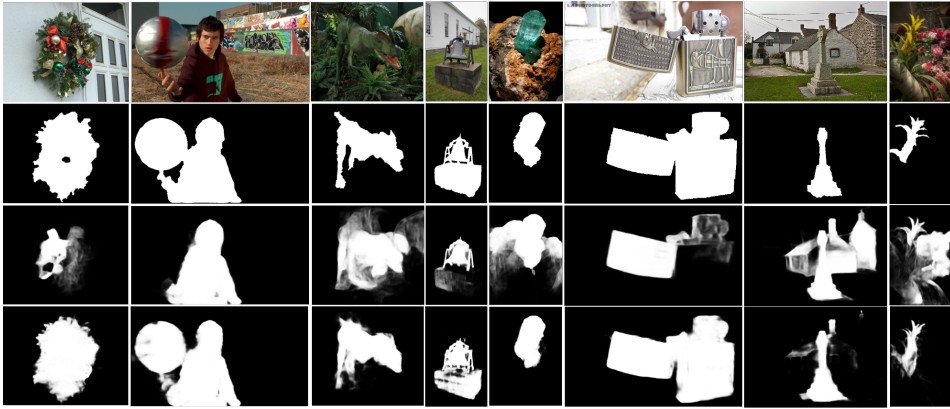

Figure 3: Visual comparison of our model and the state-of-the-art saliency prediction model, the BBSNet [16]. From top to bottom: images, ground truth saliency maps, results of the BBSNet [16] and results obtained by our model.

inference seeks to optimize $\min_\beta \min_\phi \mathrm{KL}(p_{\mathrm{data}}(s|\mathbf{I})q_\phi(z|s,\mathbf{I})||p_\beta(z,s|\mathbf{I}))$, which can be further decomposed into $\min_\beta \min_\phi \mathrm{KL}(p_{\mathrm{data}}(s|\mathbf{I})||p_\beta(s|\mathbf{I})) + \mathrm{KL}(q_\phi(z|s,\mathbf{I})||p_\beta(z|s,\mathbf{I}))$. That is, the variational inference maximizes the conditional data likelihood plus a KL-divergence between the approximate inference network and the posterior distribution. Only when the $\mathrm{KL}(q_\phi(z|s,\mathbf{I})||p_\beta(z|s,\mathbf{I})) \to 0$, the variational inference will lead to the MLE solution, which is exactly the objective of our model. However, there might exist a gap between them in practise due to the improper design of the inference network. Our learning algorithm is much simpler and more accurate than amortized inference.

**Computational and memory costs analysis:** From the learning perspective, due to the iterative Langevin sampling for the posterior and prior distributions of latent variables, our model is more time-consuming for training than generative models with amortized inference, such as VAE, which is roughly 2.4 times faster than ours on RGB image-based salient object detection. However, for VAE, the inference model is parameterized by another set of parameters, which need to be updated by back-propagation. In our model, the Langevin dynamics is not treated as a model because once the top-down generator is updated in each iteration, the posterior distribution can be derived from the generator. With the posterior distribution, the Langevin sampling is just an optimization-like process to find fair samples in the latent space defined by the posterior distribution. Without relying on an extra inference network, our framework is efficient in memory and friendly for network design.

## 4 Experimental Results

### 4.1 Setup

**Datasets:** For RGB SOD, we train models on DUTS training set [62], and test them on five benchmark datasets, including DUTS testing set, ECSSD [82], DUT [83], HKU-IS [35] and PASCAL-S [36]. For RGB-D SOD, we follow the conventional training setting, in which the training set is a combination of 1,485 images from NJU2K dataset [29] and 700 images from NLPR dataset [50]. We test the trained models on NJU2K testing set, NLPR testing set, DES [8], SSB [46] and SIP [15] testing set.

**Evaluation Metrics:** We adopt four evaluation metrics to measure the performance, including Mean Absolute Error $\mathcal{M}$, Mean F-measure ($F_\beta$), Mean E-measure ($E_\xi$) [14] and S-measure ($S_\alpha$) [13].

**Implementation Details:** Our generative vision transformer is built upon the Swin transformer [40] and uses it as the backbone of the encoder in our framework. The Swin backbone can be initialized with the parameters pretrained on the ImageNet-1K [9] dataset for image classification, and the other parameters of the newly added components, including the decoder part and the MLP of the energy based prior model, will be randomly initialized from the Gaussian distribution $\mathcal{N}(0, 0.01)$. Empirically we set the number of dimensions $d$ of the latent variables $z$ as $d = 32$. We set $\sigma_\epsilon = 1$ in Eq. (1) and $\sigma_z = 1$ in Eq. (2). We resize all the images and the saliency maps to the resolution of $384 \times 384$ pixels to fit the Swin transformer. The maximum epoch is 50. The initial learning rates

are $2.5 \times 10^{-5}$. The whole training takes 9 hours with a batch size $n' = 10$ on one NVIDIA GTX 2080Ti GPU for each model. During testing, our model can process 15 images per second.

## 4.2 Performance Comparison

We compare our framework with the state-of-the-art RGB SOD models and RGB-D SOD models, and show a comparison of performance in Table 1 and Table 2 respectively, where VST [39] is the only transformer-based saliency detection model. We observe consistently better performance of the proposed frameworks. Especially for PASCAL-S dataset [36], which contains more than 40% examples with large-sized salient objects, the significant performance gap between our method and the existing solutions demonstrates the effectiveness of our generative framework for saliency prediction. We further show a qualitative comparison between our RGB-D saliency prediction model and the BBSNet [16] in Figure 3. As we can see, our transformer-based framework, with an effective global context modeling, is superior to its competitor in detecting various sizes of salient objects.

Table 1: Performance comparison with benchmark RGB salient object detection models.

| Method | DUTS [62] | | | | ECSSD [82] | | | | DUT [83] | | | | HKU-IS [35] | | | | PASCAL-S [36] | | | |
|---|---|---|---|---|---|---|---|---|---|---|---|---|---|---|---|---|---|---|---|---|
| | $S_\alpha \uparrow$ | $F_\beta \uparrow$ | $E_\xi \uparrow$ | $\mathcal{M} \downarrow$ | $S_\alpha \uparrow$ | $F_\beta \uparrow$ | $E_\xi \uparrow$ | $\mathcal{M} \downarrow$ | $S_\alpha \uparrow$ | $F_\beta \uparrow$ | $E_\xi \uparrow$ | $\mathcal{M} \downarrow$ | $S_\alpha \uparrow$ | $F_\beta \uparrow$ | $E_\xi \uparrow$ | $\mathcal{M} \downarrow$ | $S_\alpha \uparrow$ | $F_\beta \uparrow$ | $E_\xi \uparrow$ | $\mathcal{M} \downarrow$ |
| CPD [67] | .869 | .821 | .898 | .043 | .913 | .909 | .937 | .040 | .825 | .742 | .847 | .056 | .906 | .892 | .938 | .034 | .848 | .819 | .882 | .071 |
| SCRN [68] | .885 | .833 | .900 | .040 | .920 | .910 | .933 | .041 | .837 | .749 | .847 | .056 | .916 | .894 | .935 | .034 | .869 | .833 | .892 | .063 |
| PoolNet [37] | .887 | .840 | .910 | .037 | .919 | .913 | .938 | .038 | .831 | .748 | .848 | .054 | .919 | .903 | .945 | .030 | .865 | .835 | .896 | .065 |
| BASNet [52] | .876 | .823 | .896 | .048 | .910 | .913 | .938 | .040 | .836 | .767 | .865 | .057 | .909 | .903 | .943 | .032 | .838 | .818 | .879 | .076 |
| EGNet [94] | .878 | .824 | .898 | .043 | .914 | .906 | .933 | .043 | .840 | .755 | .855 | .054 | .917 | .900 | .943 | .031 | .852 | .823 | .881 | .074 |
| F3Net [64] | .888 | .852 | .920 | .035 | .919 | .921 | .943 | .036 | .839 | .766 | .864 | .053 | .917 | .910 | .952 | .028 | .861 | .835 | .898 | .062 |
| ITSD [98] | .886 | .841 | .917 | .039 | .920 | .916 | .943 | .037 | .842 | .767 | .867 | .056 | .921 | .906 | .950 | .030 | .860 | .830 | .894 | .066 |
| SCNet [88] | .902 | .870 | .936 | .032 | .928 | .930 | .955 | .030 | .847 | .778 | .879 | .053 | .927 | .917 | .960 | .026 | .873 | .846 | .909 | .058 |
| LDF [65] | .892 | .861 | .925 | .034 | .919 | .923 | .943 | .036 | .839 | .770 | .865 | .052 | .920 | .913 | .953 | .028 | .860 | .856 | .901 | .063 |
| VST [39] | .896 | .842 | .918 | .037 | .932 | .911 | .943 | .034 | .850 | .771 | .869 | .058 | .928 | .903 | .950 | .030 | .873 | .832 | .900 | .067 |
| **Ours** | **.908** | **.875** | **.942** | **.029** | **.935** | **.935** | **.962** | **.026** | **.858** | **.797** | **.892** | **.051** | **.930** | **.922** | **.964** | **.023** | **.877** | **.855** | **.915** | **.054** |

Table 2: Performance comparison with benchmark RGB-D salient object detection models.

| Method | NJU2K [29] | | | | SSB [46] | | | | DES [8] | | | | NLPR [50] | | | | SIP [15] | | | |
|---|---|---|---|---|---|---|---|---|---|---|---|---|---|---|---|---|---|---|---|---|
| | $S_\alpha \uparrow$ | $F_\beta \uparrow$ | $E_\xi \uparrow$ | $\mathcal{M} \downarrow$ | $S_\alpha \uparrow$ | $F_\beta \uparrow$ | $E_\xi \uparrow$ | $\mathcal{M} \downarrow$ | $S_\alpha \uparrow$ | $F_\beta \uparrow$ | $E_\xi \uparrow$ | $\mathcal{M} \downarrow$ | $S_\alpha \uparrow$ | $F_\beta \uparrow$ | $E_\xi \uparrow$ | $\mathcal{M} \downarrow$ | $S_\alpha \uparrow$ | $F_\beta \uparrow$ | $E_\xi \uparrow$ | $\mathcal{M} \downarrow$ |
| BBSNet [16] | .921 | .902 | .938 | .035 | .908 | .883 | .928 | .041 | .933 | .910 | .949 | .021 | .930 | .896 | .950 | .023 | .879 | .868 | .906 | .055 |
| BiaNet [92] | .915 | .903 | .934 | .039 | .904 | .879 | .926 | .043 | .931 | .910 | .948 | .021 | .925 | .894 | .948 | .024 | .883 | .873 | .913 | .052 |
| CoNet [28] | .911 | .903 | .944 | .036 | .896 | .877 | .939 | .040 | .906 | .880 | .939 | .026 | .900 | .859 | .937 | .030 | .868 | .855 | .915 | .054 |
| UCNet [85] | .897 | .886 | .930 | .043 | .903 | .884 | .938 | .039 | .934 | .919 | .967 | .019 | .920 | .891 | .951 | .025 | .875 | .867 | .914 | .051 |
| JLDCF [17] | .902 | .885 | .935 | .041 | .903 | .873 | .936 | .040 | .931 | .907 | .959 | .021 | .925 | .894 | .955 | .022 | .880 | .873 | .918 | .049 |
| DSA2F [59] | .903 | .901 | .923 | .039 | .904 | .898 | .933 | .036 | .920 | .896 | .962 | .021 | .918 | .897 | .950 | .024 | - | - | - | - |
| VST [39] | .922 | .898 | .939 | .035 | .913 | .879 | .937 | .038 | .943 | .920 | .965 | .017 | .932 | .897 | .951 | .024 | .904 | .894 | .933 | .040 |
| **Ours** | **.929** | **.924** | **.956** | **.028** | **.916** | **.898** | **.950** | **.032** | **.945** | **.928** | **.971** | **.016** | **.938** | **.921** | **.966** | **.018** | **.906** | **.908** | **.940** | **.037** |

## 4.3 Backbone Analysis

To test the performance of the transformer structure proposed in Section 3.3, we compare it with a conventional convolutional backbone for salient object detection. We create the baseline by replacing the Swin encoder in our transformer structure by the ResNet50 [24] encoder and keeping the decoder part (i.e., the "Feature Aggregation" module shown in Figure 2) unchanged for a fair comparison.

We first test them without involving latent variables, which means that we need to train them in a discriminative manner. We use "Ours-Swin-D" to denote the model using the Swin encoder and "Ours-Res50-D" to denote the one using the ResNet50 encoder. Model performance in the task of RGB-D saliency detection are shown in Table 3. We observe that "Ours-Swin-D" outperforms "Ours-Res50-D", which clearly indicates the effectiveness of transformer backbone for salient object detection.

Table 3: Analysis of different backbones without involving latent variables.

| Method | NJU2K [29] | | | | SSB [46] | | | | DES [8] | | | | NLPR [50] | | | | SIP [15] | | | |
|---|---|---|---|---|---|---|---|---|---|---|---|---|---|---|---|---|---|---|---|---|
| | $S_\alpha \uparrow$ | $F_\beta \uparrow$ | $E_\xi \uparrow$ | $\mathcal{M} \downarrow$ | $S_\alpha \uparrow$ | $F_\beta \uparrow$ | $E_\xi \uparrow$ | $\mathcal{M} \downarrow$ | $S_\alpha \uparrow$ | $F_\beta \uparrow$ | $E_\xi \uparrow$ | $\mathcal{M} \downarrow$ | $S_\alpha \uparrow$ | $F_\beta \uparrow$ | $E_\xi \uparrow$ | $\mathcal{M} \downarrow$ | $S_\alpha \uparrow$ | $F_\beta \uparrow$ | $E_\xi \uparrow$ | $\mathcal{M} \downarrow$ |
| Ours-Res50-D | .908 | .896 | .931 | .036 | .906 | .889 | .927 | .037 | .918 | .907 | .948 | .022 | .921 | .893 | .951 | .024 | .874 | .856 | .917 | .049 |
| Ours-Swin-D | .919 | .923 | .947 | .032 | .914 | .897 | .943 | .033 | .931 | .919 | .959 | .022 | .933 | .912 | .951 | .022 | .897 | .899 | .931 | .041 |

Table 4: Performance of different backbones within our model for RGB saliency prediction.

| Method | DUTS [62] | | | | ECSSD [82] | | | | DUT [83] | | | | HKU-IS [35] | | | | PASCAL-S [36] | | | |
|---|---|---|---|---|---|---|---|---|---|---|---|---|---|---|---|---|---|---|---|---|
| | $S_\alpha \uparrow$ | $F_\beta \uparrow$ | $E_\xi \uparrow$ | $\mathcal{M} \downarrow$ | $S_\alpha \uparrow$ | $F_\beta \uparrow$ | $E_\xi \uparrow$ | $\mathcal{M} \downarrow$ | $S_\alpha \uparrow$ | $F_\beta \uparrow$ | $E_\xi \uparrow$ | $\mathcal{M} \downarrow$ | $S_\alpha \uparrow$ | $F_\beta \uparrow$ | $E_\xi \uparrow$ | $\mathcal{M} \downarrow$ | $S_\alpha \uparrow$ | $F_\beta \uparrow$ | $E_\xi \uparrow$ | $\mathcal{M} \downarrow$ |
| Ours-Res50 | .890 | .850 | .927 | .035 | .918 | .914 | .944 | .036 | .837 | .762 | .867 | .053 | .917 | .906 | .952 | .029 | .859 | .830 | .896 | .063 |
| Ours-DPT | .899 | .874 | .940 | .031 | .924 | .933 | .956 | .031 | .854 | .792 | .890 | .054 | .922 | .920 | .960 | .026 | .870 | .854 | .911 | .055 |
| **Ours-Swin** | **.908** | **.875** | **.942** | **.029** | **.935** | **.935** | **.962** | **.026** | **.858** | **.797** | **.892** | **.051** | **.930** | **.922** | **.964** | **.023** | **.877** | **.855** | **.915** | **.054** |

Table 5: Performance of different backbones within our model for RGB-D saliency prediction.

| Method | NJU2K [29] | | | | SSB [46] | | | | DES [8] | | | | NLPR [50] | | | | SIP [15] | | | |
|---|---|---|---|---|---|---|---|---|---|---|---|---|---|---|---|---|---|---|---|---|
| | $S_\alpha \uparrow$ | $F_\beta \uparrow$ | $E_\xi \uparrow$ | $\mathcal{M} \downarrow$ | $S_\alpha \uparrow$ | $F_\beta \uparrow$ | $E_\xi \uparrow$ | $\mathcal{M} \downarrow$ | $S_\alpha \uparrow$ | $F_\beta \uparrow$ | $E_\xi \uparrow$ | $\mathcal{M} \downarrow$ | $S_\alpha \uparrow$ | $F_\beta \uparrow$ | $E_\xi \uparrow$ | $\mathcal{M} \downarrow$ | $S_\alpha \uparrow$ | $F_\beta \uparrow$ | $E_\xi \uparrow$ | $\mathcal{M} \downarrow$ |
| Ours-Res50 | .919 | .909 | .946 | .033 | .906 | .882 | .937 | .038 | .937 | .925 | .974 | .017 | .920 | .892 | .949 | .025 | .882 | .872 | .918 | .049 |
| Ours-DPT | .924 | .913 | .950 | .031 | .915 | .892 | .946 | .034 | .941 | .921 | .968 | .017 | .935 | .913 | .964 | .019 | .901 | .903 | .933 | .038 |
| **Ours-Swin** | **.929** | **.924** | **.956** | **.028** | **.916** | **.898** | **.950** | **.032** | **.945** | **.928** | **.971** | **.016** | **.938** | **.921** | **.966** | **.018** | **.906** | **.908** | **.940** | **.037** |

We further study the influence of different encoder backbones in the context of the proposed generative saliency prediction framework with an EBM prior. Table 4 and Table 5, respectively, depict the performance comparisons of our frameworks using different backbones in the tasks of RGB and RGB-D salient object detections. We compare the ResNet50 encoder backbone and the Swin transformer encoder backbone. We also modify the DPT [54] backbone such that it can adapt to the tasks of RGB and RGB-D salient object detections with our framework. The DPT built on the ViT transformer [10] is originally designed for semantic segmentation and depth estimation. The comparison results verify the effectiveness of the vision transformer used in our generative framework. In particular, our current solution with the Swin transformer encoder backbone can achieve the best performance for saliency prediction. Further, the performance gap between "Ours-Swin-D" in Table 3 and "Ours-Swin" in Table 5 indicates the effectiveness of the generative learning with an expressive latent space.

## 4.4 Generative Learning Analysis

We compare our framework with other alternative generative solutions in Table 6. The results are reported in the task of RGB-D saliency detection. For fair comparison, we implement an ABP-based model [22], a GAN-based model [20], and a VAE-based model [31, 56] using the same transformer-based generator as ours. The latent variables in these models are still assumed to follow the isotropic Gaussian distribution, as in their original algorithms. To be specific, for the ABP-based model, we use MCMC-based inference while training, and sample the latent variables directly from the Gaussian distribution during testing. For the GAN-based alternative, we design a fully convolutional discriminator [26] that consists of five $3 \times 3$ convolutional layers with a stride of 2 in each layer. The discriminator takes the concatenation of an image and a saliency map as input, and is trained to distinguish between the predicted saliency map and the ground truth given an image. The numbers of the output channels of the discriminator are $64, 64, 64, 64$ and $1$. For the VAE-based alternative, we introduce an extra encoder as an approximate Gaussian inference model via the reparameterization trick. The encoder consists of four $4 \times 4$ convolutional layers with a stride of 2 in each layer and maps the concatenation of an image and a saliency map to feature maps of channel sizes $64, 64, 64$ and $64$ sequentially. After that, two fully connected layers are adopted to produce the mean and the standard deviation of the inference model. For all the three alternative generative models, we set the numbers of the dimension of the latent space the same as that in our model, which is $d = 32$.

Table 6: Performance of different generative models with transformer backbones for SOD.

| Method | NJU2K [29] | | | | SSB [46] | | | | DES [8] | | | | NLPR [50] | | | | SIP [15] | | | |
|---|---|---|---|---|---|---|---|---|---|---|---|---|---|---|---|---|---|---|---|---|
| | $S_\alpha \uparrow$ | $F_\beta \uparrow$ | $E_\xi \uparrow$ | $\mathcal{M} \downarrow$ | $S_\alpha \uparrow$ | $F_\beta \uparrow$ | $E_\xi \uparrow$ | $\mathcal{M} \downarrow$ | $S_\alpha \uparrow$ | $F_\beta \uparrow$ | $E_\xi \uparrow$ | $\mathcal{M} \downarrow$ | $S_\alpha \uparrow$ | $F_\beta \uparrow$ | $E_\xi \uparrow$ | $\mathcal{M} \downarrow$ | $S_\alpha \uparrow$ | $F_\beta \uparrow$ | $E_\xi \uparrow$ | $\mathcal{M} \downarrow$ |
| ABP | .920 | .915 | .951 | .030 | .910 | .890 | .942 | .035 | .935 | .920 | .962 | .018 | .930 | .914 | .962 | .020 | .900 | .898 | .935 | .039 |
| GAN | .928 | .922 | .954 | .030 | .913 | .892 | .941 | .033 | .940 | .924 | .969 | .018 | .934 | .915 | .961 | .021 | .901 | .904 | .937 | .039 |
| VAE | .928 | .921 | .955 | .029 | .914 | .894 | .947 | .033 | .942 | .922 | .970 | .017 | .934 | .914 | .961 | .020 | .904 | .906 | .935 | .038 |
| **Ours** | **.929** | **.924** | **.956** | **.028** | **.916** | **.898** | **.950** | **.032** | **.945** | **.928** | **.971** | **.016** | **.938** | **.921** | **.966** | **.018** | **.906** | **.908** | **.940** | **.037** |

As shown in Table 6, compared with the deterministic baseline "Ours-Swin-D" (in Table 3), the three generative frameworks in achieve better or comparable performance. Especially in the DES dataset [8], they achieve large performance improvements. Our model outperforms all these alternative generative solutions, showing the effectiveness of the informative EBM prior distribution used in our model.

## 4.5 Hyperparameter Analysis

The main hyperparameters in our framework include the number of Langevin steps $K$, the Langevin step size $\delta$, the number of dimensions of the latent space $d$, and the size of EBM prior $C_e$. We have two sets of hyperparameters $\{\delta^-, K^-\}$ and $\{\delta^+, K^+\}$ of the Langevin dynamics for sampling from the prior distribution and the posterior distribution, respectively. As to the Langevin step size, we find stable model performance with $\delta^- \in [0.2, 0.6]$ and $\delta^+ \in [0.05, 0.3]$, and we set $\delta^- = 0.4$ and $\delta^+ = 0.1$ in our paper. For the number of Langevin steps, we empirically set $K^- = K^+ = 5$ to achieve a trade-off between the training efficiency and the model performance, as more Langevin steps will lead to longer training time but more convergent inference results. Additionally, we investigate the influence of the number of latent dimensions by varying $d = \{8, 16, 32, 64\}$, and observe comparable performance among different choices of $d$. We set $d = 32$ in our paper. We also investigate the influence of the EBM size by varying $C_e = \{20, 60, 100\}$, and show the results in Table 7, in which we find $C_e = 60$ can provide optimal saliency prediction performance.

Table 7: Influence of the size of the EBM prior model

| Method | NJU2K [29] | | | | SSB [46] | | | | DES [8] | | | | NLPR [50] | | | | SIP [15] | | | |
|---|---|---|---|---|---|---|---|---|---|---|---|---|---|---|---|---|---|---|---|---|
| | $S_\alpha \uparrow$ | $F_\beta \uparrow$ | $E_\xi \uparrow$ | $\mathcal{M} \downarrow$ | $S_\alpha \uparrow$ | $F_\beta \uparrow$ | $E_\xi \uparrow$ | $\mathcal{M} \downarrow$ | $S_\alpha \uparrow$ | $F_\beta \uparrow$ | $E_\xi \uparrow$ | $\mathcal{M} \downarrow$ | $S_\alpha \uparrow$ | $F_\beta \uparrow$ | $E_\xi \uparrow$ | $\mathcal{M} \downarrow$ | $S_\alpha \uparrow$ | $F_\beta \uparrow$ | $E_\xi \uparrow$ | $\mathcal{M} \downarrow$ |
| $C_e = 20$ | **.930** | .913 | .957 | **.026** | **.917** | .898 | .950 | .030 | **.951** | .921 | .970 | **.016** | .937 | .913 | .950 | .022 | .901 | .892 | .931 | .037 |
| $C_e = 100$ | .926 | .904 | .950 | .031 | .916 | .891 | **.952** | .031 | .942 | **.930** | .970 | **.016** | **.939** | .913 | .960 | .021 | .903 | .895 | .934 | **.036** |
| **Ours** ($C_e = 60$) | .929 | **.924** | **.956** | .028 | .916 | **.898** | .950 | .032 | .945 | .928 | **.971** | **.016** | .938 | **.921** | **.966** | **.018** | **.906** | **.908** | **.940** | .037 |

## 4.6 Explainability Analysis

As a generative model, our framework is capable of obtaining a meaningful uncertainty map that summarizes the stochastic behavior of the model in performing saliency prediction. The uncertainty in our paper refers to the stochastic property of generating the saliency prediction from an input image, which is captured by the probabilistic model $p_\beta(s|\mathbf{I})$. In general, a generative saliency prediction method can provide not only accurate predictions but also reasonable uncertainty maps that represent the "subjective nature" of the human visual saliency. In this section, we propose to use the uncertainty map to help qualitatively evaluate different generative saliency prediction frameworks. Specifically, we propose to compute the uncertainty map as the variance of multiple saliency predictions produced from the learned probabilistic model. In the experiment, for each input image, we first output ten saliency maps by using the Langevin sampling from the learned conditional distribution, and then compute the variance map (uncertainty) based on the generated saliency predictions.

To quantify the complexity of a color image, we calculate a complexity score by the following way: (i) we first over-segment the image with the SLIC [1] to obtain 200 superpixels, (ii) then we compute the similarity of each superpixel with the others using handcrafted features from [99], which gives us a 200-dimensional contrast vector, representing an overall contrast of the image, (iii) we define the complexity of the image as the mean entropy of the contrast vector. The higher the score, the more complicated the image. In general, a reasonable generative saliency prediction model should have more confident predictions on images with simpler backgrounds (i.e., lower complexity scores), and less confident predictions on images with complicated backgrounds (i.e., higher complexity scores).

In Figure 4, we compare the uncertainty maps of different generative models that we have presented in Section 4.4. For each row of Figure 4, we display an input image, which is tagged with an image complexity score, and the associated ground truth saliency map, followed by the predicted saliency maps and the uncertainty maps obtained by the ABP-based model, the GAN-based model, the VAE-based model, and our model, respectively. As shown in Figure 4, the complexity score of each image is reasonable in the sense that it is consistent with the human perception. We observe that the uncertainty maps obtained by the other generative models fail to reflect the difficulties of the images for saliency prediction. For example, the image shown in the first row of Figure 4 has a structured foreground (i.e., a temple) and a textured background (i.e., a forest), which leads to a large ambiguity of the boundary between the foreground and the background. The uncertainty map obtained by our model indicates a big variance around the boundary region, which demonstrates the explainability and the reasonability of our model. Note that only generative saliency prediction frameworks can provide such an explainability analysis based on the uncertainty maps. This might motivate us to develop generative frameworks for explainable saliency prediction in the future.

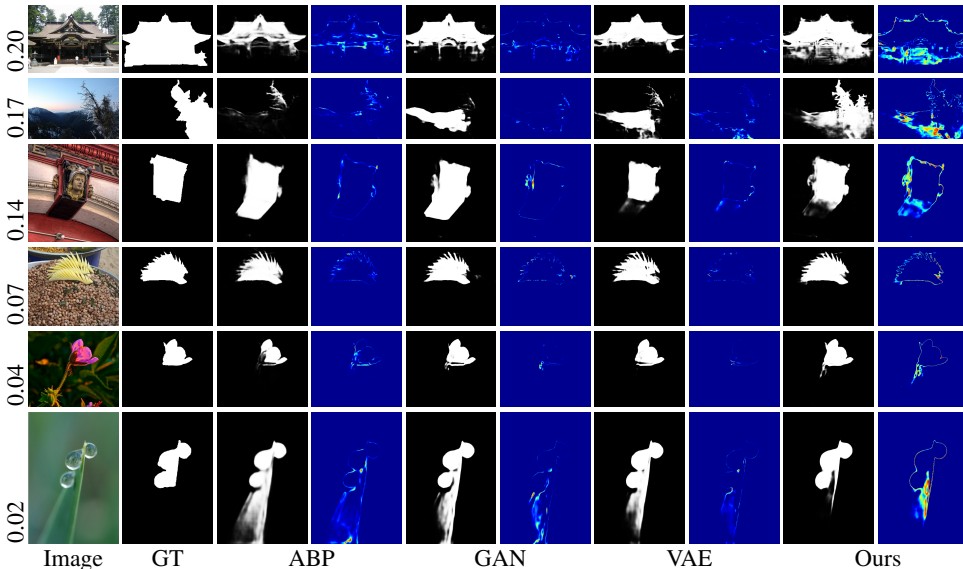

Figure 4: A comparison of uncertainty maps obtained by different generative saliency prediction frameworks for explainability analysis. Each row represents one example, in which we display an image tagged with a complexity score, the corresponding ground truth saliency map, as well as the predicted saliency maps and the uncertainty maps obtained by different generative frameworks.

# 5   Conclusion and Discussion

In this paper, we study the generative modeling and learning of vision transformer in the context of RGB and RGB-D salient object detections. We start from defining a conditional probability distribution of saliency map given an input image by a top-down latent variable generative framework, in which the non-linear mapping from image domain to saliency domain is parameterized by a proposed vision transformer network and the prior distribution of the low-dimensional latent space is represented by a trainable energy-based model. Instead of using amortized inference and sampling strategies, we learn the model by the MCMC-based maximum likelihood, where the Langevin sampling is used to evaluate the intractable posterior and prior distributions of the latent variables for calculating the learning gradients of the model parameters. With the informative energy-based prior and the expressive top-down vision transformer network, our model can achieve both accurate predictions and meaningful uncertainty maps that are consistent with the human perception.

From the machine learning perspective, our model is a likelihood-based top-down deep conditional generative model, which is neither CGAN-based nor CVAE-based frameworks, and it is trained without relying on any assisting network. The learning algorithm derived from the proposed model is based on MCMC inference for the posterior and MCMC sampling for the prior, which makes our framework more natural, principled, and statistically rigorous than others. The MCMC-based inference is immediately available in the sense that there is nothing to worry about the non-trivial design and training of a separate inference model as in VAEs. Such a framework is not only useful for saliency prediction (Though this is what we target in this paper) but also applicable to a vast of conditional learning scenarios, such as semantic segmentation, image-to-image translation, etc. Thus, the proposed generative model and the learning algorithm are generic and universal.

From the computer vision perspective, our model with a special network design to handle saliency prediction is a new member of the family of saliency prediction methods. In comparison with the traditional discriminative saliency prediction methods, our generative method is natural and reasonable because it models the saliency prediction as a conditional probability distribution. Moreover, it demonstrates impressive performance over all RGB and RGB-D SOD benchmarks. As we know, the computer vision community has started to develop vision transformer networks for various computer vision tasks, such as image classification, segmentation, detection, generation, etc. Our paper is the first one to present a generative vision transformer for both RGB and RGB-D saliency predictions. Thus, the proposed framework is significantly important for the computer vision community.

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
