# OpenReview forum: "Learning Generative Vision Transformer with Energy-Based Latent Space for Saliency Prediction"
_NeurIPS.cc/2021/Conference — NeurIPS 2021 Poster_

### Official Review · Reviewer_xC44 · 2021-07-12

**Rating:** 6
**Confidence:** 5

**Summary:**

The author construct a generative model for salient object detection in the form of top-down conditional latent variable model.
A generative vision transformer is used by adding latent variables into the traditional deterministic transformer. Experimental results on RGB and RGB-D image salient object detection show that the generative framework equipped with the EBM prior and the transformer-based non-linear mapping is powerful in representing the conditional distribution of object saliency given an image.

**Ethical Concerns:**

No ethical conerns

**Limitations And Societal Impact:**

The suggestions are presented in Main Review.

**Main Review:**

This paper exhibits a generative vision transformer for salient object detection. The idea using generative vision transformer to address the salient object detection is creative. The extensive experiments also show that it achieves the SOTA performance.

My concerns and suggestions about this paper is that many details are missing.
1. Figure 1 presents the uncertainty map, but there is no introduction presented as it is important in this paper. what is the uncertainty map, how to get it?
2. In line 51, EBM occurs at the first time, but there is no introduction about what is EBM in the whole paper.
3. In line 122, ''energy-based prior, which is presented in Eq.(1)'', which one term is the energy based prior? In addition, in some places, energy-based prior is mentioned or energy-based prior model. Does the two mean the same?
4. In line 162-166, the energy-based prior model is introduced, but it is not shown in Fig2. How does the transformer and the prior model work together? it should be explained in more detail.
5. For equation 7 and 8, it uses z_t and z_(t+1), but in the algorithm 1, step 4, 5, z_{t}^{-} and z_{t}^{+} are used. It's ambiguous to differentiate them.
6. In line 194-198, computation and memory cost are introduced. But there is no detailed comparison given in the paper. I think the comparison between the proposed and previous SOTA for computation time and memory should be detailed.
7. The performance comparison should include transformer based method, like recent work "P2T: Pyramid Pooling Transformer
for Scene Understanding".


**Time Spent Reviewing:**

10

---

> ### Author Response · Authors · 2021-08-10
> **Response to reviewer xC44**
>
> We thank the reviewer for pointing out that the proposed method in our paper is creative and the performance is state-of-the-art. We will address your concerns below and follow your suggestions to add more details to improve our paper. We hope you can reconsider your rating after your reading our replies. Thanks.
>
> **Q1. Fig. 1 presents uncertainty map without explaining what is uncertainty map and how to get it.**
>
> We will add the definition and the computation of the uncertainty map in our revision. The generative model-based saliency prediction framework provides stochastic saliency predictions given an input image, representing "subjective nature" or "uncertainty" of visual saliency. In our paper, we propose to compute this uncertainty map as the "variance of multiple saliency predictions" produced from the learned probabilistic model $p_{\beta}(s|\textbf{I})$, where $s$ is a saliency map, and $\textbf{I}$ is an image. In the experiment, for each input image, we output 10 saliency maps by using Langevin sampling from the learned conditional distribution, and compute the variance map (uncertainty) based on the predictions. We will clarify the uncertainty computation in the revision.
>
> **Q2. When EBM appears for the first time in the paper, no introduction about it is presented.**
>
> We will replace it by "energy-based model (EBM)'' for the first use of the acronym "EBM'' in our revision. Thanks.
>
> **Q3. Which is energy based prior in Eq. (1)? Do the energy-based prior model and energy-based prior mean the same?**
>
> The energy-based prior in Eq.(1) is $z \sim p_\alpha(z)$, where $p_{\alpha}(z)$ is defined in equation (2). People in statistics usually call the distribution of the latent variables as the "prior'' or "prior model''. Therefore, the "energy-based prior'' is the same as "energy-based prior model'' in our paper. We will make it clear in the revised paper. Sorry for the confusion and thanks for your carefulness.
>
> **Q4. In line 162-166, the energy-based prior model is introduced, but not shown in Fig. 2. How does the transformer and the EBM work together?**
>
> In line 162-166, we only introduced the model network architecture of the energy-based prior model. The mathematical formula of the energy-based prior model is given in equation (2) and the learning gradient of the energy-based prior model is derived in equation (9). We will edit Fig. 2 to include the energy-based prior model for better illustration. The Algorithm 1 has presented a full description of the training of the transformer and the EBM prior. The key idea is to compute the learning gradients of the parameters $\theta$ of the transformer using equation (10) and the learning gradients of the parameters $\alpha$ of the energy-based prior using equation (9), so that we can update the parameters by the Adam optimizer. Since both equation (9) and (10) involve MCMC samples $z^+$ from the posterior and $z^-$ from the prior, the whole learning algorithm is an EM-like framework, which alternates the MCMC sampling step (i.e., lines 4 and 5 in Algorithm 1) and the parameter update step (i.e., lines 6 and 7 in Algorithm 1). All these procedures are derived from the maximum likelihood estimation of the whole proposed model. Thus, it is statistically rigorous. Detailed implementation can be seen in supp Fig. 7 and Fig. 8 with training code. We will follow your suggestions to improve the paper by adding more details and reorganize the contents. Thanks.
>
> **Q5. In Eq.7,8, the latent variable z is ambiguous to differentiate.**
>
> We will revise Eq. (7) by using $z^{-}$ and Eq. (8) by using $z^{+}$ to make them more consistent with the algorithm 1. Thank you for your valuable suggestion.
>
>
> **Q6. Computation and memory cost of proposed method and SOTA methods should be detailed.**
>
> We will add computation and memory cost comparison in our revision. Please see Table 2 and Table 3 below for detailed comparison, where "M" is short for million and testing time ("Test") is measured by frame-per-second.
>
> Table 1: Computation cost of RGB SOD Models.
>
> | Method | LDF [1] | ITSD [2] | EGNet [3] | Ours |
> |---|---|---|---|---|
> | Memory | 75M | 73M | 61M | 86M |
> | Test  |  8 | 10  |  15 |  12 |
>
> Table 2: Computation cost of RGB-D SOD Models.
>
> | Method | BBSNet [4] | DSA2F [5] | JLDCF [6] | Ours |
> |---|---|---|---|---|
> | Memory | 89M | 78M | 133M | 86M |
> | Test  |  17 | 14  |  8 |  12 |
>
> [1] Label Decoupling Framework for Salient Object Detection. CVPR 2020
>
> [2] Interactive Two-Stream Decoder for Accurate and Fast Saliency Detection. CVPR 2020
>
> [3] EGNet: Edge Guidance Network for Salient Object Detection. ICCV 2019
>
> [4] BBSNet: RGB-D Salient Object Detection with a Bifurcated Backbone Strategy Network. ECCV 2020
>
> [5] Deep RGB-D Saliency Detection with Depth-Sensitive Attention and Automatic Multi-Modal Fusion. CVPR 2021
>
> [6] JL-DCF: Joint Learning and Densely-Cooperative Fusion Framework for RGB-D Salient Object Detection. CVPR 2020
>
> **Q7. Performance comparison with transformer based method should be included.**
>
> We will follow your suggestion and add the following comparison (see Table 3 and Table 4) of transformer-based methods, including the transformer-based scene understanding paper (i.e., P2T [7]) you mentioned.
>
> Because the work of P2T [7] is **released after NeurIPS submission deadline** and the code is not available, we only compare the saliency results published in their paper (i.e., three testing datasets with mean absolute error) in Table 3 and Table 4, where "S", "F", "E" and "M" represent S-measure, mean F-measure, mean E-measure and MAE respectively.
>
> Also, we show performance of using another transformer backbone, namely PVT [8] for saliency detection. Similarly, as no code is available, we only compare with their published results.
>
> Further, we show performance of using DPT [9] (**a newly accepted ICCV 2021 transformer-based dense prediction model** for semantic segmentation and depth estimation after NeurIPS deadline) as transformer backbones for saliency detection. As its code is available, we can adapt DPT for salient object detection. Besides, we also show performance of "VST" [10], which is **a newly accepted ICCV 2021 transformer-based saliency detection paper** (after NeurIPS deadline). We can see that our method outperforms other transformer-based frameworks, which further justify the superiority of our solution. Thanks for your valuable suggestion to help improve our paper.
>
> [7] P2T: Pyramid Pooling Transformer for Scene Understanding. June 2021, Arxiv.
>
> [8] Pyramid Vision Transformer: A Versatile Backbone for Dense Prediction without Convolutions. February 2021, Arxiv.
>
> [9] Vision Transformers for Dense Prediction. July 2021, ICCV.
>
> [10] Visual Saliency Transformer. July 2021, ICCV.
>
> Table 3: Performance comparison with transformer-based RGB SOD frameworks.
>
> |||DUTS||||ECSSD||||DUT||||HKU-IS||||PASCAL||||SOD|||
> |----------|-------|-------|-------|-------|-------|-------|-------|-------|-------|-------|-------|-------|--------|-------|-------|-------|----------|-------|-------|-------|-------|-------|-------|-------|
> |Method|S$\uparrow$|F$\uparrow$|E$\uparrow$|M$\downarrow$|S$\uparrow$|F$\uparrow$|E$\uparrow$|M$\downarrow$|S$\uparrow$|F$\uparrow$|E$\uparrow$|M$\downarrow$|S$\uparrow$|F$\uparrow$|E$\uparrow$|M$\downarrow$|S$\uparrow$|F$\uparrow$|E$\uparrow$|M$\downarrow$|S$\uparrow$|F$\uparrow$|E$\uparrow$|M$\downarrow$|
> PVT[8]  |. |. |. |.032 |. |. |. |. |. |. |. |.050 |. |. |. |. |. |. |. |.060 |. |. |. |. |
> P2T[7]  |. |. |. |.029 |. |. |. |. |. |. |. |.045 |. |. |. |. |. |. |. |.053 |. |. |. |. |
> DPT[9]  |.899 |.874 |.944 |.031 |.924 |.933 |.956 |.031 |.854 |.800 |.890 |.054 |.922 |.922 |.965 |.026 |.870 |.864 |.911 |.055 |.844 |.850 |.880 |.071 |
>   VST[10]  |.896 |.842 |.918 |.037 |.932 |.911 |.943 |.034 |.850 |.771 |.869 |.058 |.928 |.903 |.950 |.030 |.873 |.832 |.900 |.067 |.854 |.833 |.879 |.065 |
> Ours\_RGB |**.912** |**.891** |**.951** |**.025** |**.936** |**.940** |**.964** |**.025** |**.858** |**.802** |**.892** |**.044** |**.928** |**.926** |**.966** |**.023** |**.874** |**.876** |**.918** |**.053** |**.850** |**.855** |**.886** |**.064**  |
>
> Table 4: Performance comparison with transformer-based RGB-D SOD frameworks.
>
> |||NJU2K||||SSB||||DES||||NLPR||||LFSD||||SIP|||
> |----------|-------|-------|-------|-------|-------|-------|-------|-------|-------|-------|-------|-------|--------|-------|-------|-------|----------|-------|-------|-------|-------|-------|-------|-------|
> |Method|S$\uparrow$|F$\uparrow$|E$\uparrow$|M$\downarrow$|S$\uparrow$|F$\uparrow$|E$\uparrow$|M$\downarrow$|S$\uparrow$|F$\uparrow$|E$\uparrow$|M$\downarrow$|S$\uparrow$|F$\uparrow$|E$\uparrow$|M$\downarrow$|S$\uparrow$|F$\uparrow$|E$\uparrow$|M$\downarrow$|S$\uparrow$|F$\uparrow$|E$\uparrow$|M$\downarrow$|
> DPT[9]  |.900 |.890 |.929 |.046 |.898 |.879 |.929 |.045 |.928 |.918 |.957 |.022 |.912 |.886 |.945 |.029 |.863 |.846 |.888 |.073 |.875 |.866 |.905 |.057 |
>    VST[10]  |.922 |.898 |.939 |.035 |.913 |.879 |.937 |.038 |.943 |.920 |.965 |.017 |.932 |.897 |.951 |.024 |.882 |.871 |.917 |.061 |.904 |.894 |.933 |.040 |
>    Ours\_RGBD |**.932** |**.927** |**.959** |**.026** |**.921** |**.905** |**.953** |**.030** |**.947** |**.940** |**.979** |**.014** |**.938** |**.922** |**.966** |**.019** |**.889** |**.876** |**.920** |**.052** |**.907** |**.913** |**.943** |**.035**  |

---

> > ### Comment · Reviewer_xC44 · 2021-08-24
> > **Response to Author**
> >
> > The author basically answered my questions. But for the question Q7, the author does not list the f-measure for P2T, PVT. Actually, on the DUTS dataset, the F-measure for PVT is 90.0 and the F-measure for P2T is 91.2. From this view, it cannot show any improvement for the proposed method.
> >
> > Though the author replied that the P2T is published after the submit. But the paper doesn't present the comparison between using the proposed method and swin-encoder with simple decoder like P2T. Similar to Reviewer d7vK, No baseline shows the advantage of proposed method and author's reply doesn't solve this question. I think it's important. So I insist on my score.

---

> > > ### Author Response · Authors · 2021-08-24
> > > **Response**
> > >
> > > Thank you for your reply.
> > >
> > > Firstly, the 90.0 and 91.2 are max F-measure for PVT and P2T on DUTS testing dataset, and our reported F-measure is mean F-measure. Our max F-measure is 93.4.
> > >
> > > Secondly, our decoder is already very simple, with 1M extra parameters.
> > >
> > > Thirdly, we introduce a new generative model based saliency detection framework with both saliency prediction and reliable uncertainty map. The effectiveness of both outputs have been extensively explained in the main paper (Table 1, 2, 3, 4 and Figure 6)  and the supplementary material (Figure 10).

---

> > > > ### Comment · Reviewer_xC44 · 2021-08-30
> > > > **Response To the Authors**
> > > >
> > > > I have a last question left for this paper. How about the performance for the swin-encoder + decoder? Why the author didn't report this? I think this is important issue for d7vK and me.
> > > >
> > > > In Table 6 and 7, it just shows the swin-encoder based method outperforms others. i think the author should give a more fair comparison. For example, the performance for different decoder (used in P2T, the proposed in this paper), then different encoders with the same decoder (proposed in this paper).

---

> > > > > ### Author Response · Authors · 2021-08-31
> > > > > **Reply to your follow-up question regarding architecture design of encoder and decoder**
> > > > >
> > > > > Thank you for your response. Your questions are always welcome. We are happy to report an ablation study regarding different choices of encoder and decoder in our generative framework. Actually, the current structure, i.e., swin-encoder + ConvNet-decoder, used in our paper is determined after we conducted a comparison of performances of different combinations of encoder and decoder. Please see the following Tables.
> > > > >
> > > > >
> > > > > To be specific, Table 1 summarizes a comparison of different choices of encoders, where the decoder is a simple ConvNet structure that is used in our paper. The candidates for the encoder include ResNet50, VGG16, and swin. **From Table 1, we can see that using a Transformer (i.e., swin) as encoder can lead to much better performance**.
> > > > >
> > > > > Table 2 summarizes a comparison of different choices of decoders, where the encoder is a swin structure that is used in our paper. The candidates for the decoder include swin, ConvNet, and P2T's decoder.
> > > > >
> > > > > As there is no public code available for P2T, we re-implement the decoder for the visual saliency P2T based framework as described. Especially, we re-implement a SCPC block [1] according to the introduction in [1] and attach it after each backbone feature as in P2T. Then, we use the U-Net decoder in [2] (same as the decoder for saliency detection in P2T) for lower-higher level feature aggregation.
> > > > >
> > > > > **From Table 2, we find that the model using a simple ConvNet as decoder performs comparably to the one with a swin decoder. However, the former uses much less parameters than the latter, thus our ConvNet decoder is more competitive. (Note that: the numbers of parameters (i.e., size) of swin decoder, P2T decoder, and ours are 36M, 78M and 1M, respectively.)**
> > > > >
> > > > > We will include the above comparisons as an ablation study for the structural design of our model in the revision. Thank you for your valuable suggestions to improve our paper. We appreciate it. If you have further questions, please feel free to let us know.
> > > > >
> > > > >
> > > > > [1] EDN: Salient object detection via extremely-downsampled network. Arxiv 2020
> > > > >
> > > > > [2] U-Net: Convolutional networks for biomedical image segmentation. MICCAI 2015
> > > > >
> > > > >
> > > > >
> > > > > **Table 1: A comparison of performances of different choices of encoders (with our ConvNet-decoder)**
> > > > >
> > > > > |||NJU2K||||SSB||||DES||||NLPR||||LFSD||||SIP|||
> > > > > |----------|-------|-------|-------|-------|-------|-------|-------|-------|-------|-------|-------|-------|--------|-------|-------|-------|----------|-------|-------|-------|-------|-------|-------|-------|
> > > > > |(encoder+decoder) |S$\uparrow$|F$\uparrow$|E$\uparrow$|M$\downarrow$|S$\uparrow$|F$\uparrow$|E$\uparrow$|M$\downarrow$|S$\uparrow$|F$\uparrow$|E$\uparrow$|M$\downarrow$|S$\uparrow$|F$\uparrow$|E$\uparrow$|M$\downarrow$|S$\uparrow$|F$\uparrow$|E$\uparrow$|M$\downarrow$|S$\uparrow$|F$\uparrow$|E$\uparrow$|M$\downarrow$|
> > > > > ResNet50+ConvNet |.911 |.899 |.939 |.037 |.892 |.870 |.926 |.044 |.927 |.908 |.963 |.023 |.920 |.891 |.949 |.024 |.835 |.823 |.867 |.089 |.895 |.890 |.926 |.046  |
> > > > > | VGG16+ConvNet  | .921        | .910        | .946        | .033        | .909        | .886        | .937        | .037        | .926    | .913        | .960        | .021        | .925    | .897        | .952        | .024        | .855        | .834        | .881        | .077        | .888        | .876        | .923        | .047          |
> > > > > | Swin+ConvNet(Ours) | **.932** | **.927** | **.959** | **.026** | **.921** | **.905** | **.953** | **.030** | **.947** | **.940** | **.979** | **.014** | **.938** | **.922** | **.966** | **.019** | **.889** | **.876** | **.920** | **.052** | **.907** | **.913** | **.943** | **.035**
> > > > >
> > > > >
> > > > >
> > > > > **Table 2: A comparison of performances of different choices of decoders (with a swin-encoder)**
> > > > >
> > > > > |||NJU2K||||SSB||||DES||||NLPR||||LFSD||||SIP|||
> > > > > |----------|-------|-------|-------|-------|-------|-------|-------|-------|-------|-------|-------|-------|--------|-------|-------|-------|----------|-------|-------|-------|-------|-------|-------|-------|
> > > > > |(encoder+decoder)|S$\uparrow$|F$\uparrow$|E$\uparrow$|M$\downarrow$|S$\uparrow$|F$\uparrow$|E$\uparrow$|M$\downarrow$|S$\uparrow$|F$\uparrow$|E$\uparrow$|M$\downarrow$|S$\uparrow$|F$\uparrow$|E$\uparrow$|M$\downarrow$|S$\uparrow$|F$\uparrow$|E$\uparrow$|M$\downarrow$|S$\uparrow$|F$\uparrow$|E$\uparrow$|M$\downarrow$|
> > > > > |Swin+Swin|**.937** |**.928** |**.960** |**.026** |**.925** |.902 |**.956** |.031 |**.948** |.935 |.978 |**.014** |.936 |.913 |.964 |**.018** |.884 |.863 |.912 |.053 |**.907** |.903 |**.944** |**.035** |
> > > > > | Swin+P2T   | .921        | .902        | .954        | .030        | .916        | .884        | .954        | .035        | .934    | .907        | .967        | .016        | .923    | .892        | .957        | .020        | .861        | .850        | .899        | .068        | .890        | .880        | .928        | .041          |
> > > > > | Swin+ConvNet(Ours) | .932 | .927 | .959 | **.026** | .921 | **.905** | .953 | **.030** | .947 | **.940** | **.979** | **.014** | **.938** | **.922** | **.966** | .019 | **.889** | **.876** | **.920** | **.052** | **.907** | **.913** | .943 | **.035**

---

> > > > > > ### Author Response · Authors · 2021-09-03
> > > > > > **We appreciate your insightful questions and we are looking forward to your final decision**
> > > > > >
> > > > > > Dear Reviewer xC44,
> > > > > >
> > > > > > We hope our last reply with an additional ablation study can make you feel satisfied to accept our work.
> > > > > >
> > > > > > Your insightful questions have motivated us to conduct additional experiments to substantially strengthen our paper.
> > > > > >
> > > > > > We are so happy to see we have get them done during the rebuttal. We appreciate your contributions. You are our hero  !!!!
> > > > > >
> > > > > > We are looking forward to your final decision!
> > > > > >
> > > > > > best,
> > > > > >
> > > > > > Authors

---

> > > > > > ### Comment · Reviewer_xC44 · 2021-09-03
> > > > > > **Response to author**
> > > > > >
> > > > > > My questions and concerns are solved based on the author's detailed reply. Thus i re-scored this paper.

---

> > > ### Author Response · Authors · 2021-08-25
> > > **Our method outperforms P2T, PVT, and other baselines**
> > >
> > > Thank you for your reply. The following are our responses to your further questions and concerns.
> > >
> > > **Q1: About comparison with baselines P2T [7] and PVT [8]**
> > >
> > > I think you made a mistake in compraring our method with P2T [7] and PVT [8] because you are comparing their **max F-measure** with our **mean F-measure**. Actually, our method outperforms P2T and PVT in terms of MAE and max F-measure.  Please see the following detailed explanation.
> > >
> > > In Table 7 of the P2T paper [7], they reported the model performance on three testing dataset: DUTS testing dataset, DUT (or DUT-O datset) and PASCAL saliency testing dataset. They use two metrics: **Max F-measure** and MAE. While, in our paper, we report **Mean F-measure** and MAE. (In our rebuttal, both Table 3 and Table 4 above also report **mean F-measure** and MAE). As we have no access to their codes, we could only compare our method with their reported MAE in Table 3 and Table 4 above.
> > >
> > > Now, for a fair comparison, we can also compare our method with P2T and PVT in terms of max F-measure. Table 5 shows a comparsion of our method, P2T, and PVT in terms of **max F-measure** and MAE. Our method outperforms P2T and PTV in terms of **max F-measure** and MAE. The significant performance improvements indicate the advantage of the proposed framework. We will add these results to our revision. Please re-consider your rating.
> > >
> > >
> > >
> > > Table 5: Performance comparison with two transformer backbone based models.
> > >
> > > ||DUTS||DUT||PASCAL||
> > > |----------|-------|-------|-------|-------|-------|-------|
> > > |Method|F$_\max\uparrow$|MAE$\downarrow$|F$_\max\uparrow$|MAE$\downarrow$ |F$_\max\uparrow$|MAE$\downarrow$|
> > > PVT[8]  |.900|.032 |.832 |.050 |.883 |.060 |
> > > P2T[7]  |.912|.029 |.840 |.045 |.898 |**.053**  |
> > > Ours\_RGB |**.934** |**.025** |**.866** |**.044** |**.917** |**.053** |
> > >
> > >
> > > [7] P2T: Pyramid Pooling Transformer for Scene Understanding. June 2021, Arxiv.
> > >
> > > [8] Pyramid Vision Transformer: A Versatile Backbone for Dense Prediction without Convolutions. February 2021, Arxiv.
> > >
> > >
> > >
> > > **Q2: No baseline shows the advantage of proposed method**
> > >
> > > In total, we have further compared with two convolutional backbone based models ([11] and [12]) and transformer based saliency detection models ([7,8,9,10]) as shown in Table 6 and Table 7 (also Table 5 as above), which we believe is sufficient to demonstrate the superior performance of our framework. Also, we show the baseline models with non-informative Gaussian prior or model without prior in Table 8. We can see that the effectiveness of using EBM prior is obvious. Models using an EBM prior outperform models without prior or using a non-informative Gaussian prior. Among models with EBM priors using different sizes of energy functions, we find C=60 can provide optimal prediction performance. The comparison is fair, because all the comparing methods in Table 8 use the same network structures except the difference in the prior setting.**Note that, all these three tables regarding baseline comparison are also discussed in Reviewer d7vK's response section.** We will add these results to our revision. Please re-consider your rating accordingly.
> > >
> > > [9] Vision Transformers for Dense Prediction. July 2021, ICCV.
> > >
> > > [10] Visual Saliency Transformer. July 2021, ICCV.
> > >
> > > [11] LDF: Label Decoupling Framework for Salient Object Detection. CVPR 2020
> > >
> > > [12] DAS2F: Deep RGB-D Saliency Detection with Depth-Sensitive Attention and Automatic Multi-Modal Fusion. CVPR 2021
> > >
> > > Table 6: RGB Saliency Performance Comparison.
> > >
> > > |||DUTS||||ECSSD||||DUT||||HKU-IS||||PASCAL||||SOD|||
> > > |----------|-------|-------|-------|-------|-------|-------|-------|-------|-------|-------|-------|-------|--------|-------|-------|-------|----------|-------|-------|-------|-------|-------|-------|-------|
> > > |Method|S$\uparrow$|F$\uparrow$|E$\uparrow$|M$\downarrow$|S$\uparrow$|F$\uparrow$|E$\uparrow$|M$\downarrow$|S$\uparrow$|F$\uparrow$|E$\uparrow$|M$\downarrow$|S$\uparrow$|F$\uparrow$|E$\uparrow$|M$\downarrow$|S$\uparrow$|F$\uparrow$|E$\uparrow$|M$\downarrow$|S$\uparrow$|F$\uparrow$|E$\uparrow$|M$\downarrow$|
> > > LDF[11]  |.892 |.855 |.910 |.034 |.918 |.930 |.925 |.034 |.839 |.773 |.873 |.051 |.919 |.914 |.954 |.027 |.863 |.848 |.864 |.060 |- |- |- |-  |
> > > DPT[9]  |.899 |.874 |.944 |.031 |.924 |.933 |.956 |.031 |.854 |.800 |.890 |.054 |.922 |.922 |.965 |.026 |.870 |.864 |.911 |.055 |.844 |.850 |.880 |.071 |
> > >   VST[10]  |.896 |.842 |.918 |.037 |.932 |.911 |.943 |.034 |.850 |.771 |.869 |.058 |.928 |.903 |.950 |.030 |.873 |.832 |.900 |.067 |**.854** |.833 |.879 |.065 |
> > >    Ours\_RGB |**.912** |**.891** |**.951** |**.025** |**.936** |**.940** |**.964** |**.025** |**.858** |**.802** |**.892** |**.044** |**.928** |**.926** |**.966** |**.023** |**.874** |**.876** |**.918** |**.053** |.850 |**.855** |**.886** |**.064**  |
> > >
> > > Table 7: RGB-D Saliency Performance Comparison.
> > >
> > > |||NJU2K||||SSB||||DES||||NLPR||||LFSD||||SIP|||
> > > |----------|-------|-------|-------|-------|-------|-------|-------|-------|-------|-------|-------|-------|--------|-------|-------|-------|----------|-------|-------|-------|-------|-------|-------|-------|
> > > |Method|S$\uparrow$|F$\uparrow$|E$\uparrow$|M$\downarrow$|S$\uparrow$|F$\uparrow$|E$\uparrow$|M$\downarrow$|S$\uparrow$|F$\uparrow$|E$\uparrow$|M$\downarrow$|S$\uparrow$|F$\uparrow$|E$\uparrow$|M$\downarrow$|S$\uparrow$|F$\uparrow$|E$\uparrow$|M$\downarrow$|S$\uparrow$|F$\uparrow$|E$\uparrow$|M$\downarrow$|
> > > DSA2F[12]  |.903 |.901 |.923 |.039 |.904 |.898 |.933 |.036 |.920 |.896 |.962 |.021 |.918 |.897 |.950 |.024 |.882 |**.882** |**.923** |.054 |- |- |- |- |
> > > DPT[9]  |.900 |.890 |.929 |.046 |.898 |.879 |.929 |.045 |.928 |.918 |.957 |.022 |.912 |.886 |.945 |.029 |.863 |.846 |.888 |.073 |.875 |.866 |.905 |.057 |
> > >    VST[10]  |.922 |.898 |.939 |.035 |.913 |.879 |.937 |.038 |.943 |.920 |.965 |.017 |.932 |.897 |.951 |.024 |.882 |.871 |.917 |.061 |.904 |.894 |.933 |.040 |
> > > |Ours\_RGBD |**.932** |**.927** |**.959** |**.026** |**.921** |**.905** |**.953** |**.030** |**.947** |**.940** |**.979** |**.014** |**.938** |**.922** |**.966** |**.019** |**.889** |.876 |.920 |**.052** |**.907** |**.913** |**.943** |**.035** |
> > >
> > >
> > > Table 8: Prior related baseline models
> > >
> > > |||NJU2K||||SSB||||DES||||NLPR||||LFSD||||SIP|||
> > > |----------|-------|-------|-------|-------|-------|-------|-------|-------|-------|-------|-------|-------|--------|-------|-------|-------|----------|-------|-------|-------|-------|-------|-------|-------|
> > > |Method|S$\uparrow$|F$\uparrow$|E$\uparrow$|M$\downarrow$|S$\uparrow$|F$\uparrow$|E$\uparrow$|M$\downarrow$|S$\uparrow$|F$\uparrow$|E$\uparrow$|M$\downarrow$|S$\uparrow$|F$\uparrow$|E$\uparrow$|M$\downarrow$|S$\uparrow$|F$\uparrow$|E$\uparrow$|M$\downarrow$|S$\uparrow$|F$\uparrow$|E$\uparrow$|M$\downarrow$|
> > > No\_Prior |.930 |.924 |.959 |.026 |.915 |.895 |.949 |.032 |.941 |.929 |.975 |.015 |.934 |.912 |.962 |.020 |.875 |.863 |.905 |.063 |.898 |.900 |.939 |.040  |
> > > |Gaussian\_Prior|.930 |.915 |.951 |.030 |.918 |.890 |.942 |.035 |.945 |.921 |.969 |.017 |.940 |.914 |.962 |.020 |.883 |.862 |.902 |.063 |.907 |.898 |.935 |.039 |
> > > |EBM\_Prior (C=60) |**.932** |**.927** |**.959** |**.026** |**.921** |**.905** |**.953** |**.030** |.947 |**.940** |**.979** |**.014** |.938 |**.922** |**.966** |**.019** |**.889** |**.876** |**.920** |**.052** |**.907** |**.913** |**.943** |**.035** |
> > > EBM\_Prior (C=20) |.930 |.913 |.957 |.026 |.917 |.898 |.950 |.030 |**.951** |.921 |.970 |.016 |.937 |.913 |.950 |.022 |.883 |.860 |.900 |.063 |.901 |.892 |.931 |.037  |
> > > EBM\_Prior (C=100) |.926 |.904 |.950 |.031 |.916 |.891 |.952 |.031 |.942 |.930 |.970 |.016 |**.939** |.913 |.960 |.021 |.875 |.950 |.895 |.064 |.903 |.895 |.934 |.036  |

---

### Official Review · Reviewer_6hfU · 2021-07-15

**Rating:** 7
**Confidence:** 2

**Summary:**

This paper introduces a novel* supervised generative model for detecting salient image regions, where saliency is defined as containing object. This is a latent-variable model with an energy-based prior and a transformer decoder that maps the latent variable to a saliency map. The prior and posterior inference is done via SGLD. The model is well-evaluated on several datasets, with several metrics, and compared across a range of state-of-the-art baselines. It not only outperforms every single baseline (setting new SOTA), but also has the added benefit of providing high-quality uncertainty estimates. Detailed ablation studies show the relative importance of various model components.

\* I think it is novel, but difficult to say as I am not expert in the field and there is very information in the paper about prior art.

# Update
I'm happy with the author response and am increasing my score to 7. I still have some doubts about the motivation of this work, but the theory is sound and the the results are very good.

**Limitations And Societal Impact:**

Adequate discussion.

**Main Review:**

Originality:
- The task is well-established (silent object detection, or SOD for short. This is at least as far as I can tell, given the number of baselines.
- It is not clear how this work differs from prior art. Even though there are numerous baselines in the experiments, they are not discussed. The related work section lists a huge number of related works but does not discuss how these differ from the current work. Or rather, it does so only implicitly, stating only very shortly what some other methods do. Since I am not an expert in SOD, it is difficult for me to understand the differences to the current method with so little information.
- This work is a combination of well-known techniques: energy-based generative modelling for the prior, SGLD for prior and posterior inference, a transformer for decoding. It is this combination and its application to SOD that, as far as I can tell (I am not an expert in the field), is novel.

Quality:
- The presented method and results are very good quality, but not very surprising: it is clear that a generative model can provide good uncertainty estimates.
- The paper needs some writing work (more in clarity).

Good:
- The submission is technically sound. The details of the method are well-explained and seem correct.
- An extensive experimental evaluation is provided. The method outperforms numerous baselines on several datasets.
- The authors provide a short-but-sufficient analysis of the learning algorithm (with prior/posterior SGLD sampling).
- There are numerous ablation studies evaluating importance of model components.
- I really like the "hyperparameter analysis" section in L263-L276. It gathers and describes all hparams of the method and says how they were tuned and if the model was sensitive. Nice!

Bad:
- Even though the model can provide some uncertainty estimates, it cannot estimate the uncertainty in the model parameters. The authors do not comment on that.
- One of the main motivations of an EBM prior is that a correlated prior/posterior should do better than (conditionally-) independent ones as in a GAN or a VAE. While "obvious" on the surface, we have to remember that an independent prior + a single MLP layer (not to mention more) can lead to a highly-correlated prior, cf. normalizing flows. Therefore, I would say that his motivation is invalid---despite the fact that the authors instantiate GAN and VAE variants of their model and show they underperform the EBM.
- I am not convinced that Eq. 4 is fully correct. I tried deriving it, but I only managed to get the expectation of the gradient with respect to the model distribution of (latent, saliency), but training is done with ground-truth saliency and posterior-sampled latents. Could you please walk me through the derivation?
- In L198 you say that your method is more memory-efficient than e.g. VAE because there is no encoder. However, for SGLD, you need to evaluate the prior and the decoder multiple times. Do you backprop through SGLD? If yes, then you need to store activations of every prior/decoder evaluation, which renders your method *very* memory-inefficient. Care to elaborate?

Clarity:
- This submission would benefit from some more writing work. It is quite unclear in places, see comments below.
- There is a sufficient level of detail to reimplement the model, provided that the reader can understand the paper.
- There are a lot of typos, e.g. L44 none->non, L75 modal->modality, L226 & L278 analysis -> analyse.
- "SOD" is first used in L65 but not explained; I needed to infer that it means "salient object detection". Similar "saliency prediction problem" appears in L103 without definition. The paper does not state until very far into the text that it tackles supervised learning.
- The related works section is almost unreadable. It lists tens of different works without giving any differences between them, nor specifying how they relate to the current work.
- Since prior/posterior samples are defined with + and - signs, it would be nice to define the related hyperparams in Alg 1 using + and - as well (instead of 0 and 1).
- L192 "and other newly added layers randomly initialized" <- I've no idea what this refers to.
- It is very unclear that Tab. 3 lists deterministic baselines; perhaps add that in the caption? and also mention in Tab. 4 caption that a deterministic variant of "ours" is available in Tab. 3. The paragraph describing the deterministic backbone study (L227-238) is very difficult to read and the fact that the model is now deterministic is highly obscured.
- Unclear what "scales" in Fig. 4 means.
- Unclear what 0/1 as supervision in L251 means.
- Fig. 5 is difficult to read; please provide an interpretation.
- Please add a ylabel to Fig. 6.


Significance:
- While it is nice to see an application of energy-based generative modelling to supervised learning, I am not sure how significant this is. I am not an expert in this field, and I have no idea where SOD is used and what are the benefits as compared to e.g. object detection or instance/panoptic segmentation.
- Given that there is a number of SOD baselines in the paper, I can be convinced that SOD is an important research area. Since the presented results are quite a bit better than the baselines, and the method is simple, it is likely that it will be built upon.

I would be happy to increase my score, potentially quite significantly, if the authors address my concerns above and clearly delineate themselves from prior art.

**Time Spent Reviewing:**

5

---

> ### Author Response · Authors · 2021-08-10
> **Response to reviewer 6hfU-Part 1 of 2**
>
> We would like to thank the reviewer for the detailed review and valuable comments.
>
> **Q1. How the work is different from prior art.**
>
> We will discuss the novelty of our model from two perspectives. (1) A new generative models: Among all the existing image generative models, our method is the first vision-transformer-based conditional latent variable model (conditional generator) using an energy-based prior. [1] is another vision-transformer-based generative model following the GAN framework, which was posted in ArXiv Jul 2021 after the NeurIPS submission. Other existing vision-transformer-based frameworks are discriminative models, for example, [2] is a model for image recognition, [3] is for dense prediction, and [4] is for object detection. Our model belongs to vision-transformer-based generative models but it still differs from [1] because ours is trained by MCMC-based maximum likelihood estimation instead of adversarial training. Besides, our method uses an energy-based model as a prior, while [1] uses the Gaussian prior as in other traditional GAN frameworks. Our model is a new member in the family of generative models. It is new because it uses transformer and energy-based prior. (2) A new salient object detection method: In the field of salienct object detection (SOD), prior SOD models are mostly discriminative models, which learn a one-to-one non-linear mapping instead of a distribution for prediction. Very recently, some works start to work on vision transformer in the discriminative model for SOD, such as visual saliency transformer [5] in ICCV 2021. UC-NET [6] in CVPR 2020 is the first work that uses a generative model, i.e., traditional VAE, for RGB-D salient object detection. Following the conventional generative model setting, [6] defines the prior of the latent variable as Gaussian distribution. Except the method in [6], there is no other generative frameworks for SOD. Our paper is a vision-transformer-based generative model with an energy-based prior for saliency prediction.
>
> [1] Generative Adversarial Transformers. ArXiv Jul, 2021
>
> [2] An image is worth 16 x 16 words: Transformers for image recognition at scale. ICLR 2021.
>
> [3] vision transformer for dense prediction. ArXiv Mar. 2021.
>
> [4] Deformerable DETR: Deformable transformers for end-to-end object detection. ICLR 2021.
>
> [5] Visual Saliency Transformer, ICCV Jul, 2021.
>
> [6] uc-net: uncertainty inspired rgb-d saliency detection via conditional variational autoencoders. CVPR 2020.
>
> **Q2. The presented results are not surprising as generative model can provide good uncertainty.**
>
> We need to point out that almost all the traditional methods for saliency prediction are deterministic, so there is hardly any methods using generative models for saliency prediction. As far as we know, [6] is the first and the only one generative framework for salient object detection. Inspired by [6], our paper proposes a more powerful generative model for salient object detection, in which vision transformer structure and energy-based prior distribution are used for better uncertainty estimation. The results are surprising because, with an informative energy-based prior, our method outperforms other traditional generative models (including GANs and VAEs) in terms of accuracy and interpretability.
>
> **Q3. The model cannot estimate uncertainty in model parameters.**
>
> Our model only captures uncertainty of the data, since we learn a distribution of saliency $s$ given an image $\textbf{I}$, i.e., $p(s|\textbf{I};\beta)$. (Actually, all generative models, including GANs and VAEs, only focus on learning the data distribution, i.e., data uncertainty.) The uncertainty of the model corresponds to the distribution of the parameters, i.e., $p(\beta|s,\textbf{I})$, which usually happens in Bayesian statistics, where the parameters are treated as a random variable such that the model uncertainty can be estimated. However, our model is learned by maximum likelihood, which is a frequentist view. In general, maximum likelihood estimation (from the frequentist view) and the Bayesian estimation (from the Bayesian view) are two different ways to estimate parameters of models. Our model belongs to the first one. Thus, our paper is not related to model uncertainty.
>
> **Q4. One of the main motivation that a correlated prior/posterior from EBM should do better than independent ones, which is invalid.**
>
> This statement might be a misunderstanding. The top-down generators used in the original VAEs and GANs assume their latent variables follow a Gaussian distribution. This is a strong assumption. The motivation of our work using a trainable energy-based prior instead of a Gaussian prior is to avoid any strong assumption made on the latent variables of the top-down generators. The energy-based prior parameterized by MLP is flexible and powerful enough to represent any forms of distributions for the latent variables. We will let the data decide what kind of prior distribution is suitable for the latent variables, because the prior distribution is learned together with the generator by maximum likelihood estimation. In general, the energy-based model with the energy function parameterized by neural network is a more expressive and informative prior than the Gaussian, mixture of Gaussians, and Laplace distributions that are commonly used for priors in the traditional latent variable models.
>
>
> **Q5. derivation of Eq.4?**
>
> We would like to derive Eq.4 step by step below, and will add it to our supplementary material. The derivation mainly relies on the definition of the expectation, Bayes' theorem, and the derivative using chain rule. Please see below:
> $$
> \begin{equation}
> \begin{split}
> \nabla_{\beta} \log p_{\beta}(s|\mathbf{I})
> &= \frac{1}{p_{\beta}(s|\text{I})} \nabla_{\beta} p_{\beta}(s|\mathbf{I}) \\\\
> &= \frac{1}{p_{\beta}(s|\text{I})} \nabla_{\beta} \left[\int p_{\beta}(s, z|\mathbf{I})dz \right] \\\\
>     &= \int \left[\nabla_{\beta} \log p_{\beta}(s,z|\mathbf{I}) \right] \frac{p_{\beta}(s,z|\mathbf{I})}{p_{\beta}(s|\mathbf{I})} dz \\\\
>      &= \int \left[\nabla_{\beta} \log p_{\beta}(s,z|\mathbf{I}) \right] p_{\beta}(z|s,\mathbf{I})dz \\\\
>    &= E_{p_{\beta}(z|s,\textbf{I})} \left[\nabla_{\beta} \log p_{\beta}(s,z|\textbf{I}) \right] \\\\
> &= E_{p_{\beta}(z|s,\textbf{I})} \left[\nabla_{\beta} \log (p_{\alpha}(z) p_{\theta}(s|z,\textbf{I})) \right] \\\\
>       & =E_{p_{\beta}(z|s,\textbf{I})}[\nabla_{\beta} (\log p_{\alpha}(z)+  \log p_{\theta}(s|\textbf{I},z))] \\\\
> \end{split}
> \end{equation}
> $$
>
> **Q6. Elaborate why your method is memory efficient? Do you backprop through SGLD?**
>
> We do NOT backprop through SGLD, we only need to save the parameters of the top-down model (i.e., energy-based prior model and the transformer network). For VAE, the inference network is parameterized by the other set of parameters, which need to be updated by back-propagation. In our model, the SGLD is not treated as a model because when our model is updated in each iteration, the posterior distribution can be derived from the model. With the posterior distribution $p(z|s,\textbf{I})$, SGLD sampling is just an optimization-like process to find fair examples $\{z\}$ in the $z$ space. The examples $\{z\}$ will be used as numerical values (not a Tensor) in the next learning step, which means that we will not backprop through SGLD. Recall that the learning objective of the model is a direct maximum likelihood, without involving learning any other approximate model. The SGLD is just a numeral calculation step to approximate the expectation (i.e., using a sample average to approximate an intractable expectation). Therefore, our model is memory efficient. We will add more explanations and discussions in our revision to make this point clearer. Thanks.

---

> > ### Author Response · Authors · 2021-08-10
> > **Response to reviewer 6hfU-Part 2 of 2**
> >
> > **Q7. Typos.**
> >
> > We will fix the typos. Thanks.
> >
> > **Q8. The acronym "SOD''.**
> >
> > SOD means salient object detection. We will revise the paper to avoid confusion. Thank you.
> >
> > **Q9. The related work section is unreadable.**
> >
> > We will revise the related work section by giving difference between those works and specifying how they are related to our work. Thanks.
> >
> >
> > **Q10. It would be nice to define the related hyperparameters in Algorithm 1 using + and - (instead of 0 and 1).**
> >
> > We will follow your suggestion to revise the notation in Algorithm 1. Thanks.
> >
> > **Q11. what does “other newly added layers randomly initialized'' mean.**
> >
> > Sorry for the confusion. Our generator is built upon a backbone network, and those "newly added layers'' refer to the decoder part of the generator and the MLP of the energy based prior model. The backbone can be initialized with the pretrained parameters, and those newly added ones will be "randomly initialized'' from the Gaussian distribution of $N(0,0.01)$. We will clarify this in the revision.
> >
> > **Q12. The description of deterministic baseline in Table 3, 4 and Line 227-238 is difficult to read.**
> >
> > We will revise them accordingly to make them more readable.
> >
> > **Q13. Unclear what "scales'' in Fig. 4 means.**
> >
> > "scales'' in Fig. 4 means the percentage of the salient objects within the whole input image. Fig. 4 aims to explain that the transformer backbone is robust to the size of the salient objects. We will clarify it.
> >
> > **Q14. Unclear what 0/1 as supervision in L251 means.**
> >
> > This refers to the discriminator in GAN, where 1 is assigned for the real observation and 0 is assigned for the generated prediction by the generator. In our paper, we design a GAN-based SOD network for an ablation study, where the fully-convolutional discriminator is used to distinguish ground truth from and prediction generated by the conditional generator. We will revise the paper to make it much clearer.
> >
> > **Q15. provide an interpretation for Fig. 5.**
> >
> > In Fig. 5, we aim to analyse how the channel attention module effects the backbone network by visualizing the features. To do so, we compute the channel-wise mean of the backbone features and then perform minmax normalization to generate the feature visualization (Line 291-292). We observe that the channel attention module based model has relatively uniform activation within the same instance, leading to a better performance of "D\_CBase'' (with channel attention) compared with "D\_Base'' (without channel attention). We will revise caption of Fig. 5 to make it more readable. Thanks.
> >
> >
> > **Q16. Please add a ylabel to Fig.6.**
> >
> > Sure.
> >
> >
> > **Q17. About significance of salient object detection.**
> >
> > Firstly, as a **class-agnostic** object segmentation task, salient object detection (SOD) is usually used to provide useful prior knowledge about the scenario [7,8,9]. Secondly, salient object detection is context based task, which focuses on the part that attracts human attention, making it useful in explaining human visual attention. Even though the framework in our paper is proposed for SOD, the novel design of the model and the learning algorithm can be useful for other dense prediction tasks, such as semantic segmentation. We believe our paper will have a broader impact on the computer vision and machine learning communities.
> >
> > [7] Railroad Is Not a Train: Saliency As Pseudo-Pixel Supervision for Weakly Supervised Semantic Segmentation. CVPR 2021
> >
> > [8] Saliency-Guided Image Translation. CVPR 2021
> >
> > [9] Learning Saliency Propagation for Semi-Supervised Instance Segmentation. CVPR 2020

---

> > ### Comment · Reviewer_6hfU · 2021-08-24
> > **I'm happy with most of the responses: increasing my score to 7.**
> >
> > **Q2**: I recognise that most SOD models are deterministic. However, as a generative model practitioner, I must say that your VAE baseline is fundamentally flawed: when the posterior has no access to all information it requires (in this case, the saliency maps), it **cannot** learn good uncertainty as it will average out any uncertainty resulting from missing information. In that sense, UCNet is the correct way to construct such a VAE as first shown by [Kohl et. al.](https://arxiv.org/abs/1806.05034). My criticism holds: your results are hardly surprising as a **correctly-structured generative model** should be able to represent uncertainty well.
> >
> > **Q5**: How did you go from line 2 to line 3?
> >
> > **Q6**: Thanks, that's good to know!

---

> > > ### Author Response · Authors · 2021-08-25
> > > **Responses to the follow-up questions**
> > >
> > > Thank you for your feedback. We are very happy to see that our response have addressed your concerns and you have increased your rating. Please see below for the answers of your follow-up questions. We hope you can increase your confidence regarding to your current rating to support our work if you feel our replies have made you feel satisfied. Thanks again!
> > >
> > > **Q2: your VAE baseline is fundamentally flawed: when the posterior has no access to all information it requires (in this case, the saliency maps), it cannot learn good uncertainty as it will average out any uncertainty resulting from missing information. In that sense, UCNet is the correct way to construct such a VAE as first shown by Kohl et. al.. My criticism holds: your results are hardly surprising as a correctly-structured generative model should be able to represent uncertainty well.**
> > >
> > > Sorry for the confusion. Our VAE baseline is similar to UCNet (as well as the work by Kohl et al) except that our VAE baseline uses transformer structure as the backbone while the UCNet adopts a ResNet50 backbone. As shown in Figure 5 (part of our code) of the supplementary material, our inference model (i.e., a bottom-up encoder) $p(z|x,y)$ takes both an image $x$ and the corresponding saliency map $y$ as input and output the latent codes $z$. (You can check Figure 5 in the supplementary material. The "model.xy_encoder" is the inference network $p(z|x,y)$ that takes $x$ and $y$ as inputs; the "model.x_encoder" is the prior distribution $p(z|x)$ that takes the image $x$ as input.)
> > >
> > > Also, during the training of the VAE baseline, we have the KL divergence term defined between the posterior $p(z|x,y)$ and the piror $p(z|x)$, which is $KL(p(z|x,y)||p(z|x))$. In this way, our VAE baseline is as correct as UCNet. That is, it rigorously follows the conditional VAE framework. The more reliable uncertainty maps compared with alternative generative models (see Figure 6 of the manuscript, and Figure 10 of the supplementary material) illustrate effectiveness of our solution. We will make it clear in the revised version. Thank you for improving our presentation in the paper.
> > >
> > > **Q5: How did you go from line 2 to line 3?**
> > >
> > > Please see the following proof with more details.
> > > $$
> > > \begin{equation}
> > > \begin{split}
> > > \nabla_{\beta} \log p_{\beta}(s|\mathbf{I})&= \frac{1}{p_{\beta}(s|\mathbf{I})} \nabla_{\beta} p_{\beta}(s|\mathbf{I})\\\\
> > >     &= \frac{1}{p_{\beta}(s|\mathbf{I})} \nabla_{\beta} \left[\int p_{\beta}(s, z|\mathbf{I})dz \right]\\\\
> > >      &= \frac{1}{p_{\beta}(s|\mathbf{I})}  \left[\int \nabla_{\beta} p_{\beta}(s, z|\mathbf{I})dz \right]\\\\
> > >        &=   \int \left[\nabla_{\beta} p_{\beta}(s, z|\mathbf{I})\right] \frac{1}{p_{\beta}(s|\mathbf{I})} dz \\\\
> > >         &=   \int \left[ \frac{1}{p_{\beta}(s,z|\mathbf{I})} \nabla_{\beta} p_{\beta}(s, z|\mathbf{I})\right] \frac{p_{\beta}(s,z|\mathbf{I})}{p_{\beta}(s|\mathbf{I})} dz \\\\
> > >     &= \int \left[\nabla_{\beta} \log p_{\beta}(s,z|\mathbf{I}) \right] \frac{p_{\beta}(s,z|\mathbf{I})}{p_{\beta}(s|\mathbf{I})} dz \\\\
> > >     &= \int \left[\nabla_{\beta} \log p_{\beta}(s,z|\mathbf{I}) \right] \frac{p_{\beta}(s,z|\mathbf{I})}{p_{\beta}(s|\mathbf{I})} dz \\\\
> > >      &= \int \left[\nabla_{\beta} \log p_{\beta}(s,z|\mathbf{I}) \right] p_{\beta}(z|s,\mathbf{I})dz \\\\
> > > &= \mathbf{E}\_{p\_{\beta}(z|s,\mathbf{I})} \left[\nabla_{\beta} \log p_{\beta}(s,z|\mathbf{I}) \right]\\\\
> > >     &= \mathbf{E}\_{p\_{\beta}(z|s,\mathbf{I})} \left[\nabla_{\beta} \log (p_{\alpha}(z) p_{\theta}(s|z,\mathbf{I})) \right]\\\\
> > >       & =\mathbf{E}\_{p\_{\beta}(z|s,\mathbf{I})}[\nabla_{\beta} (\log p_{\alpha}(z)+  \log p_{\theta}(s|\mathbf{I},z))]\\\\
> > > \end{split}
> > > \end{equation}
> > > $$

---

### Official Review · Reviewer_d7vK · 2021-07-18

**Rating:** 5
**Confidence:** 4

**Summary:**

The submission proposes a method for saliency detection based on a generative adversary learning framework, which uses a transformer architecture to replace the backbone of the framework and is jointly trained with a Markov chain Monte Carlo-based maximum likelihood estimation to improve the performance. The performance of the proposed method is evaluated on both RGBD- and RGB-based saliency prediction datasets. The paper is generally written well and easy to follow.



**Limitations And Societal Impact:**

The motivation and contribution are not clear, and the experimental results and ablation study are not sufficient. Please check the main review part for more detailed comments.

**Main Review:**

Although the overall performance of the proposed method is promising, the motivation, the contribution, and the effectiveness of the method are not very clearly illustrated and demonstrated. The reviewer has some detailed comments below:

(i)	The motivation of using a GAN-based method for saliency detection is not that clear. Does it really outperform the convolutional-based discriminative models for the task? From the results, the reviewer also cannot see any performance comparison to show the advantages of using a GAN-based method as a baseline.

(ii)	It is not clear that if the contribution is an improvement of GAN-based model with a transformer, or a method that advances saliency detection. If the focus of the paper is a new GAN-based method with transformer, and saliency detection is just an application, the authors should illustrate more differences to existing works that using GAN with transformers.

(iii)	The effectiveness of energy-based prior distribution seems not demonstrated in the experimental results. The whole ablation study seems to be very limited.

(iv)	The authors claimed state-of-the-art performance on saliency detection. However, the comparison with previous sota methods is significantly not enough. To support that claim, the authors need to compare more methods on more challenging datasets.

(v)	For the presentation, an overall framework depiction could help the readers to understand the whole thing more easily. It is also important to elaborate more, probably with a figure as well, on how the proposed energy-based strategy can be jointly optimized with the network.

(vi)	The authors showed some visualization of the learned uncertainties, but the reviewer is not clear that why the uncertainty is critical in the method. More explanation on the motivation of the uncertainty is definitely helpful, and corresponding experimental results to show the benefits of the uncertainty are also needed.


**Time Spent Reviewing:**

3 hours

---

> ### Author Response · Authors · 2021-08-10
> **Response to reviewer d7vK-Part 1 of 2**
>
> We would like to thank the reviewer for the detailed review and valuable comments.
>
> **Clarifying before answer the questions.**
> We first appreciate your time to review our paper. However, we believe you might have a misunderstanding of the proposed framework, which may directly affect your rating on our paper. Specifically, our framework is NOT a GAN-based or adversary framework, and we only compared our proposed method with a GAN-based alternative in the ablation study section. In this paper, (i) we proposed a novel latent variable model (i.e., energy-based prior $p_{\alpha}(z)$ + a conditional vision transformer $p_{\theta}(s|\textbf{I},z)$) as a generative framework for supervised saliency prediction. Such a model has never been proposed among all VAE-based or GAN-based frameworks; (ii) we proposed to train such an interesting framework by maximum-likelihood estimation, which leads to an MCMC-based MLE training algorithm. We need to point out that the adversary framework is essentially different from ours because it is not based on likelihood. Most importantly, our framework is based on MCMC and does not rely on a discriminator like GAN or an inference model like VAE. This makes the whole model and the associated learning algorithm significantly different from all existing GAN-based and VAE-based models. Our paper is novel in terms of modeling and learning.
>
> **Q1. The motivation of using GAN-based method is not clear? Does it outperform discriminative models? No baseline to show the advantage of GAN-based method.**
>
> Please first read the first paragraph above in our response to your concerns, which is "Clarifying before answer the questions''. Our framework is not a GAN-based method, where it neither requires a discriminator nor is trained by a mini-max game. We guess your question is in regards to the motivation of using a generative model for saliency prediction. We will answer this question instead. The motivations can be summarized below: **a)** Both deterministic and generative methods are important modeling tools for representation learning, which is essential in all vision tasks. Traditional deterministic methods for saliency prediction have been widely studied for learning a one-to-one mapping from image to saliency, however, the progress of investigating generative models for saliency prediction (i.e. representing saliency prediction as a conditional probability $p(s|\textbf{I})$) is still lagging behind. Thus, our paper tries to make a contribution along this direction. **b)** There have been some good works studying generative models for saliency predictions. For example, a seminal work [1] (a CVPR 2020 best paper nomination) uses a VAE for RGB-D saliency detection. Our work is inspired by [1] and is a new generative model for saliency prediction. **c)** Given an image, the saliency output of a human is subjective, therefore, a stochastic generative model, i.e., $p(s|\textbf{I})$, is more natural than a deterministic model for representing saliency prediction. **d)** Generative models for salient object detection can provide the following advantages: (i) performance improvement (comparing "D\_CBase'' (Table 3 in the paper) with "Ours'' (Table 2 in the paper)) and (ii) uncertainty estimation of the data representing the subjective nature of the human saliency predictions (Fig. 6 in the paper).
>
> [1] UC-Net: Uncertainty Inspired RGB-D Saliency Detection via Conditional Variational Autoencoders. CVPR 2020.
>
> **Q2. Not clear that if the contribution is an improvement of GAN-based model with transformer or a method for saliency prediction.**
>
>
> Again, our model is NOT a GAN-based model. The first contribution is to propose a new conditional latent variable model with a vision transformer and an energy-based prior, as well as the corresponding likelihood-based training algorithm without relying on any extra assisting network. The second controbution is that we make this new framework to become a new salient object detection method, which means the design of the network architecture is speficially for the purpose of saliency prediction. We make contributions to the field of saliency prediction by proposing a new generative saliency model. At the end, we need to point out that the proposed new framework (i.e., the novel model together with the interesting learning algorithm) is not only useful for saliency prediction but also can be adapted to other dense prediction problems. We believe our paper will have a broader impact in the fields of generative models from machine learning community and saliency prediction from computer vision community. We will revise our paper to make our constributions much clearer.
>
>
>
>
> **Q3. The effectiveness of EBM as prior estimation is not well demonstrated. The ablation study is limited.**
>
> The ablation study of the effectiveness of EBM prior has been shown in Table 5 of the paper. We will revise the paper to make it much clearer latter. We now **re-organize** the existing results in our paper (see Table 1 below) to demonstrate the effectiveness of the EBM prior. We compare different model settings, including (i) a model without prior (i.e., a deterministic model without latent variables), (ii) a model with Gaussian prior, (iii) models using an EBM prior with the energy function parameterized by a 4-layer MLP, in which the last layer output dimension C is set to be 20, 60, 100 respetively. (note that the final output of the MLP energy function is a summation of all elements in the last layer. Therefore we use C to specify the size of the energy function of the EBM prior).
>
> We can see that the effectiveness of using EBM prior is obvious. Models using EBM prior outperform models without prior or using non-informative Gaussian prior. Among models with EBM prors with different sizes of energy functions, we find C=60 can provide optimal prediction perforance. The comparsion is fair, because all the comparing methods in Table 1 use the same newtwork strutures except the difference in the prior setting.
>
>
> Table 1: Ablation study of prior.
>
> |||NJU2K||||SSB||||DES||||NLPR||||LFSD||||SIP|||
> |----------|-------|-------|-------|-------|-------|-------|-------|-------|-------|-------|-------|-------|--------|-------|-------|-------|----------|-------|-------|-------|-------|-------|-------|-------|
> |Method|S$\uparrow$|F$\uparrow$|E$\uparrow$|M$\downarrow$|S$\uparrow$|F$\uparrow$|E$\uparrow$|M$\downarrow$|S$\uparrow$|F$\uparrow$|E$\uparrow$|M$\downarrow$|S$\uparrow$|F$\uparrow$|E$\uparrow$|M$\downarrow$|S$\uparrow$|F$\uparrow$|E$\uparrow$|M$\downarrow$|S$\uparrow$|F$\uparrow$|E$\uparrow$|M$\downarrow$|
> No\_Prior |.930 |.924 |.959 |.026 |.915 |.895 |.949 |.032 |.941 |.929 |.975 |.015 |.934 |.912 |.962 |.020 |.875 |.863 |.905 |.063 |.898 |.900 |.939 |.040  |
> |Gaussian\_Prior|.930 |.915 |.951 |.030 |.918 |.890 |.942 |.035 |.945 |.921 |.969 |.017 |.940 |.914 |.962 |.020 |.883 |.862 |.902 |.063 |.907 |.898 |.935 |.039 |
> |EBM\_Prior (C=60) |**.932** |**.927** |**.959** |**.026** |**.921** |**.905** |**.953** |**.030** |.947 |**.940** |**.979** |**.014** |.938 |**.922** |**.966** |**.019** |**.889** |**.876** |**.920** |**.052** |**.907** |**.913** |**.943** |**.035** |
> EBM\_Prior (C=20) |.930 |.913 |.957 |.026 |.917 |.898 |.950 |.030 |**.951** |.921 |.970 |.016 |.937 |.913 |.950 |.022 |.883 |.860 |.900 |.063 |.901 |.892 |.931 |.037  |
> EBM\_Prior (C=100) |.926 |.904 |.950 |.031 |.916 |.891 |.952 |.031 |.942 |.930 |.970 |.016 |**.939** |.913 |.960 |.021 |.875 |.950 |.895 |.064 |.903 |.895 |.934 |.036  |

---

> > ### Author Response · Authors · 2021-08-10
> > **Response to reviewer d7vK-Part 2 of 2**
> >
> >
> >
> > **Q4. Performance comparison is not enough, more challenging datasets and methods should be compared.**
> >
> > As to datasets, we have tested our method on mainstream and popular RGB and RGB-D benchmark datasets (as shown in Table 1 and Table 2 in the paper), following the convention in most salient object detection papers. We will test on more datasets in the revised paper.
> >
> > Further, we add a comparison with four extra models in Table 2 below for RGB saliency performance comparison and Table 3 below for RGB-D saliency performance comparison, where "LDF''[2] is a state-of-the-art (SOTA) RGB saliency detection model, "DAS2F''[3] is a SOTA RGB-D saliency detection model and  "DPT''[4] is newly accpeted ICCV 2021 paper using transformer for semantic segmentation and depth estimation, "VST''[5] is a new ICCV 2021 paper using discriminative transformer for salient object detection. Note that, we re-train "DPT''[4] for both RGB salient object detection and RGB-D salient object detection with the same training datasets as our models. The better performances of our models compared with both transformer-based and CNN-based models verify the effectiveness of our framework. We will add this comparison to our revised paper.
> >
> > [2] LDF: Label Decoupling Framework for Salient Object Detection. CVPR 2020
> >
> > [3] DAS2F: Deep RGB-D Saliency Detection with Depth-Sensitive Attention and Automatic Multi-Modal Fusion. CVPR 2021
> >
> > [4] Vision Transformers for Dense Prediction. July 2021, ICCV.
> >
> > [5] Visual Saliency Transformer. July 2021, ICCV.
> >
> > Table 2: RGB Saliency Performance Comparison.
> >
> > |||DUTS||||ECSSD||||DUT||||HKU-IS||||PASCAL||||SOD|||
> > |----------|-------|-------|-------|-------|-------|-------|-------|-------|-------|-------|-------|-------|--------|-------|-------|-------|----------|-------|-------|-------|-------|-------|-------|-------|
> > |Method|S$\uparrow$|F$\uparrow$|E$\uparrow$|M$\downarrow$|S$\uparrow$|F$\uparrow$|E$\uparrow$|M$\downarrow$|S$\uparrow$|F$\uparrow$|E$\uparrow$|M$\downarrow$|S$\uparrow$|F$\uparrow$|E$\uparrow$|M$\downarrow$|S$\uparrow$|F$\uparrow$|E$\uparrow$|M$\downarrow$|S$\uparrow$|F$\uparrow$|E$\uparrow$|M$\downarrow$|
> > LDF[2]  |.892 |.855 |.910 |.034 |.918 |.930 |.925 |.034 |.839 |.773 |.873 |.051 |.919 |.914 |.954 |.027 |.863 |.848 |.864 |.060 |- |- |- |-  |
> > DPT[4]  |.899 |.874 |.944 |.031 |.924 |.933 |.956 |.031 |.854 |.800 |.890 |.054 |.922 |.922 |.965 |.026 |.870 |.864 |.911 |.055 |.844 |.850 |.880 |.071 |
> >   VST[5]  |.896 |.842 |.918 |.037 |.932 |.911 |.943 |.034 |.850 |.771 |.869 |.058 |.928 |.903 |.950 |.030 |.873 |.832 |.900 |.067 |**.854** |.833 |.879 |.065 |
> >    Ours\_RGB |**.912** |**.891** |**.951** |**.025** |**.936** |**.940** |**.964** |**.025** |**.858** |**.802** |**.892** |**.044** |**.928** |**.926** |**.966** |**.023** |**.874** |**.876** |**.918** |**.053** |.850 |**.855** |**.886** |**.064**  |
> >
> > Table 3: RGB-D Saliency Performance Comparison.
> >
> > |||NJU2K||||SSB||||DES||||NLPR||||LFSD||||SIP|||
> > |----------|-------|-------|-------|-------|-------|-------|-------|-------|-------|-------|-------|-------|--------|-------|-------|-------|----------|-------|-------|-------|-------|-------|-------|-------|
> > |Method|S$\uparrow$|F$\uparrow$|E$\uparrow$|M$\downarrow$|S$\uparrow$|F$\uparrow$|E$\uparrow$|M$\downarrow$|S$\uparrow$|F$\uparrow$|E$\uparrow$|M$\downarrow$|S$\uparrow$|F$\uparrow$|E$\uparrow$|M$\downarrow$|S$\uparrow$|F$\uparrow$|E$\uparrow$|M$\downarrow$|S$\uparrow$|F$\uparrow$|E$\uparrow$|M$\downarrow$|
> > DSA2F[3]  |.903 |.901 |.923 |.039 |.904 |.898 |.933 |.036 |.920 |.896 |.962 |.021 |.918 |.897 |.950 |.024 |.882 |**.882** |**.923** |.054 |- |- |- |- |
> > DPT[4]  |.900 |.890 |.929 |.046 |.898 |.879 |.929 |.045 |.928 |.918 |.957 |.022 |.912 |.886 |.945 |.029 |.863 |.846 |.888 |.073 |.875 |.866 |.905 |.057 |
> >    VST[5]  |.922 |.898 |.939 |.035 |.913 |.879 |.937 |.038 |.943 |.920 |.965 |.017 |.932 |.897 |.951 |.024 |.882 |.871 |.917 |.061 |.904 |.894 |.933 |.040 |
> > |Ours\_RGBD |**.932** |**.927** |**.959** |**.026** |**.921** |**.905** |**.953** |**.030** |**.947** |**.940** |**.979** |**.014** |**.938** |**.922** |**.966** |**.019** |**.889** |.876 |.920 |**.052** |**.907** |**.913** |**.943** |**.035** |
> >
> > **Q5. Better illustration on how the EBM strategy can be jointly optimized with the network should be given.**
> >
> > Thanks for your valuable suggestion. We will follow your suggestion to add an illustration (i.e., a figure) to better depict the proposed training framework in our revision.
> >
> > **Q6. Why uncertainty is critical in this method? Results showing the benefits of uncertainty are needed.**
> >
> > Firstly, the uncertainty of data in our paper refers to the stochastic property of the saliency prediction given an image, which is captured by the generative saliency prediction model $p(s|\textbf{I})$. This property is interesting and natural in the saliency prediction by human being. Only generative frameworks (because they are in the form of probability distributions) can be used to estimate such an uncertainty from the data. The traditional saliency prediction methods using deterministic models can not capture this property so that they can not uncover the uncertainty of human saliency prediction hidden in the data. Secondly, in our paper, we use the uncertainty map (see Figure 6 in our paper) to help qualitatively  evaluate different generative saliency prediction methods, because a better generative saliency prediction method can provide much more interpretable and reasonable uncertainty maps. For example, compared to other generative baselines, the uncertainty maps produced by our method are more consistent with the complexity of the input images. Our paper is the first to use this property to evaluate the generative saliency prediction models. We will revise the paper to highlight the benifit and importance of the uncertainty map in saliency prediction.

---

> > > ### Author Response · Authors · 2021-08-31
> > > **looking forward to your feedback**
> > >
> > > Dear reviewer d7vK,
> > >
> > > Could you please tell us if our responses have addressed your concerns or clarified your confusion or misunderstanding?
> > >
> > > Also, if you have other questions, please feel free to let us know, we are more than happy to answer.
> > >
> > > Since the discussion period is near the end, we are looking forward to your feedback and engagement.
> > >
> > > Thanks a lot,
> > >
> > > authors

---

### Official Review · Reviewer_Re5H · 2021-07-22

**Rating:** 7
**Confidence:** 4

**Summary:**

This paper proposes a method based on generative vision transformer with latent variables
following an informative energy-based prior for salient object detection.
Both the vision transformer network and the energy based prior model are jointly trained
via Markov chain Monte Carlo (MCMC)-based maximum likelihood estimation.

Extensive experiments reveal both high performance model and meaningful confidence map scores,
as well as estimation of uncertainty maps.

**Limitations And Societal Impact:**


How do you validate the estimated uncertainty maps, in the absence of GTs for the same ?

Wonder what is the expansion of the acronym "D_CBase"  used ? What does it mean ?

What is MAE in fig. 4 ? Not defined anywhere in main doc. - got it in Supple though.

Fig. 9 of Supple material - how do you validate these complexity scores ? Even the GT or human evaluation is not provided.
What are the typical ranges of complexity values you get by the entropy based measure ?
How do you thus claim it to be consistent with human visual perception ?

Some typos:
line 169: ......local minimum, it solve the following estimating.... - change to :
it solves the....

Some more references may be added as:
1. Bayesian Learning via Stochastic Gradient Langevin Dynamics; Max Welling, Yee Whye Teh, ICML 2011;

2. Variance Reduction in Stochastic Gradient Langevin Dynamics; Avinava Dubey, Sashank J. Reddi, Barnabas P´oczos, Alexander J. Smola, Eric P. Xing, NIPS 2016.





**Main Review:**


Not assuming the prior of the GAN to be Gaussian, and instead using an energy based NL function is a good start in the model.
Most the analytics is meaningful and well drafted.

The proposed model of learning by MCMC-based maximum likelihood where the Langevin sampling is used to evaluate the intractable posterior and prior distributions of the latent variables for calculating the gradients of the log-likelihood w.r.t. the model parameters, is quite interesting. The authors however do not claim it to be novel, wondering if a variant exists somewhere in literature ?

The tables 1-5 with performance measures, may also be put as bar-charts in Supple material, as its often hard to compare enhancement of your performance wrt nearest SOA, from decimal places in mind.


**Time Spent Reviewing:**

10-12 Hrs

---

> ### Author Response · Authors · 2021-08-10
> **Response to Reviewer Re5H**
>
> We thank the reviewer for the very detailed review and valuable comments. We have provided a point-dy-point response below and will follow your suggestions to revise our paper.
>
> **Q1. Langevin sampling for posterior and prior distribution evaluation of the latent variable for log-likelihood gradients calculation is interesting, why not claim it as novelty?**
>
> Actually, we have claimed that such a joint training framework of the vision transformer and the energy-based prior model via an MCMC-based maximum likelihood estimation is one of our major contributions in line 59. (That is our 2nd contribution listed in our introduction section). We will highlight it in our revision to clarify the confusion.
>
> **Q2. Performance in Table 1-5 may be put as bar-charts for clear reference.**
>
> Thank you for your suggestion. We will include the bar-charts in the revised paper.
>
> **Q3. How to validate the uncertainty maps without GTs?**
>
> There is no way to directly validate the uncertainty maps due to the lack of GTs. In this paper, we indirectly validate the uncertainty maps with the complexity of the image, which is quantified by the complexity score. This score is based on the calculation of entropy that measures the randomness (complexity) of the image pattern. The higher the score, the more complex the input image, and the more uncertain the uncertainty map.
>
>
>
> **Q4. What is "D\_CBase"?**
>
> "D\_CBase" refers to a traditional deterministic learning baseline for supervised saliency prediction. In this baseline, we neither have the prior model nor latent variables. We will add more explanation in our revision for the term "D\_CBase" to make it more readable.
>
>
> **Q5. What is MAE in Fig. 4? It's only defined in Supp.**
>
> MAE is short for Mean Absolute Error, and we will introduce it in the revised paper.
>
> **Q6. How to validate complexity score with no GT or human evaluation provided?**
>
> In both the manuscript (Line 301) the supp (Line 48), we introduced the way to compute the complexity score of an image, which is mainly based on an assumption that a simple image usually has compact and high-contrast foreground objects, leading to a low mean image entropy (note that: entropy represents the level of randomness in information theory). We only use this score to qualitatively evaluate the uncertainty maps produced by generative models.
>
> As the proposed "complexity score" is obtained based on the randomness of the image pattern, it can roughly indicate the complexity of the image, which might affect the saliency prediction. As shown in Fig. 6 of the manuscript and Fig. 9 of the supp, we can see that the scores are reasonable and consistent with our visual perception.
>
> Since thre is no GT, it is very hard to validate the score rigorously. Again, in our paper, we only validate it with our human perception to see if it is consistent with our understanding about the image. At least so far, the score works well and can serve as a metric to check if the uncertainty map is reasonable or not. We believe this will be an important direction to explore for evaluating generative saliency prediction methods in the future.
>
> **Q7. What's the range of complexity value? How to make sure it's consistent with human visual perception?**
>
> The complexity measure $C_m$ is defined as the mean entropy of image contrast $I_m$, which is $Cm=-Im*\log(I_m)$. As we normalize $I_m$ to make it in the range of [0,1], the resulting $C_m$ is in the range of (0,0.5).
>
> We compute complexity score using superpixel-level image contrast information, which is based on an assumption that an image with lower contrast foreground objects is more complex than an image with higher contrast foreground objects. In this way, our complexity score is designed by following human visual perception, so that it should be consistent with it. Figure 6 shown in our paper has validated the consistency, for example, an image with an obvisouly complex object is associated with a high complex score. We will work on designing more effective image complexity measure in the future. Thanks.
>
> **Q8. Typos and missing references.**
>
> We will fix those typos and cite the mentioned references regarding Langevin dynamics. Thanks.

---

> > ### Author Response · Authors · 2021-08-10
> > **Adding more experimental results in our revision**
> >
> > Requested by Reviewer xC44, we will revise our paper by adding a comparson with more newly released transfomer-based saliency prediction methods (Note that: those methods are totally different from ours) and a detailed comparison on computation and memory costs. Please see our response to Q6 and Q7 raised by Reviewer xC44 for more information. We believe those newly added experimental results will further strengthen our paper and increase the contribution.

---

> > > ### Author Response · Authors · 2021-08-28
> > > **New contribution!!  A summary of the comparison with 6 new baselines**
> > >
> > > We are very excited to let you know that we have conducted extra experiments to compare our model with new baselines. The results show that our method outperforms these comparing baselines.
> > >
> > > For your convenience, we summarize the comparison below. We have further compared with two convolutional backbone-based models ([1] and [2]) and four transformer-based saliency detection models ([3,4,5,6]) as shown in Tables A, B and C, respectively. We will add these results to our revision to further strengthen our paper, even though some of them appeared after the NeurIPS submission date.
> > >
> > >
> > > [1] LDF: Label Decoupling Framework for Salient Object Detection. CVPR 2020
> > >
> > > [2] DAS2F: Deep RGB-D Saliency Detection with Depth-Sensitive Attention and Automatic Multi-Modal Fusion. CVPR 2021
> > >
> > > [3] Vision Transformers for Dense Prediction. July 2021, ICCV.
> > >
> > > [4] Visual Saliency Transformer. July 2021, ICCV.
> > >
> > > [5] Pyramid Vision Transformer: A Versatile Backbone for Dense Prediction without Convolutions. February 2021, Arxiv.
> > >
> > > [6] P2T: Pyramid Pooling Transformer for Scene Understanding. June 2021, Arxiv.
> > >
> > >
> > > Table A: RGB Saliency Performance Comparison.
> > >
> > > |||DUTS||||ECSSD||||DUT||||HKU-IS||||PASCAL||||SOD|||
> > > |----------|-------|-------|-------|-------|-------|-------|-------|-------|-------|-------|-------|-------|--------|-------|-------|-------|----------|-------|-------|-------|-------|-------|-------|-------|
> > > |Method|S$\uparrow$|F$\uparrow$|E$\uparrow$|M$\downarrow$|S$\uparrow$|F$\uparrow$|E$\uparrow$|M$\downarrow$|S$\uparrow$|F$\uparrow$|E$\uparrow$|M$\downarrow$|S$\uparrow$|F$\uparrow$|E$\uparrow$|M$\downarrow$|S$\uparrow$|F$\uparrow$|E$\uparrow$|M$\downarrow$|S$\uparrow$|F$\uparrow$|E$\uparrow$|M$\downarrow$|
> > > LDF[1]  |.892 |.855 |.910 |.034 |.918 |.930 |.925 |.034 |.839 |.773 |.873 |.051 |.919 |.914 |.954 |.027 |.863 |.848 |.864 |.060 |- |- |- |-  |
> > > DPT[3]  |.899 |.874 |.944 |.031 |.924 |.933 |.956 |.031 |.854 |.800 |.890 |.054 |.922 |.922 |.965 |.026 |.870 |.864 |.911 |.055 |.844 |.850 |.880 |.071 |
> > >   VST[4]  |.896 |.842 |.918 |.037 |.932 |.911 |.943 |.034 |.850 |.771 |.869 |.058 |.928 |.903 |.950 |.030 |.873 |.832 |.900 |.067 |**.854** |.833 |.879 |.065 |
> > >    Ours\_RGB |**.912** |**.891** |**.951** |**.025** |**.936** |**.940** |**.964** |**.025** |**.858** |**.802** |**.892** |**.044** |**.928** |**.926** |**.966** |**.023** |**.874** |**.876** |**.918** |**.053** |.850 |**.855** |**.886** |**.064**  |
> > >
> > >
> > > Table B: RGB-D Saliency Performance Comparison.
> > >
> > > |||NJU2K||||SSB||||DES||||NLPR||||LFSD||||SIP|||
> > > |----------|-------|-------|-------|-------|-------|-------|-------|-------|-------|-------|-------|-------|--------|-------|-------|-------|----------|-------|-------|-------|-------|-------|-------|-------|
> > > |Method|S$\uparrow$|F$\uparrow$|E$\uparrow$|M$\downarrow$|S$\uparrow$|F$\uparrow$|E$\uparrow$|M$\downarrow$|S$\uparrow$|F$\uparrow$|E$\uparrow$|M$\downarrow$|S$\uparrow$|F$\uparrow$|E$\uparrow$|M$\downarrow$|S$\uparrow$|F$\uparrow$|E$\uparrow$|M$\downarrow$|S$\uparrow$|F$\uparrow$|E$\uparrow$|M$\downarrow$|
> > > DSA2F[2]  |.903 |.901 |.923 |.039 |.904 |.898 |.933 |.036 |.920 |.896 |.962 |.021 |.918 |.897 |.950 |.024 |.882 |**.882** |**.923** |.054 |- |- |- |- |
> > > DPT[3]  |.900 |.890 |.929 |.046 |.898 |.879 |.929 |.045 |.928 |.918 |.957 |.022 |.912 |.886 |.945 |.029 |.863 |.846 |.888 |.073 |.875 |.866 |.905 |.057 |
> > >    VST[4]  |.922 |.898 |.939 |.035 |.913 |.879 |.937 |.038 |.943 |.920 |.965 |.017 |.932 |.897 |.951 |.024 |.882 |.871 |.917 |.061 |.904 |.894 |.933 |.040 |
> > > |Ours\_RGBD |**.932** |**.927** |**.959** |**.026** |**.921** |**.905** |**.953** |**.030** |**.947** |**.940** |**.979** |**.014** |**.938** |**.922** |**.966** |**.019** |**.889** |.876 |.920 |**.052** |**.907** |**.913** |**.943** |**.035** |
> > >
> > >
> > > Table C: Performance comparison with two transformer backbone based models.
> > >
> > > ||DUTS||DUT||PASCAL||
> > > |----------|-------|-------|-------|-------|-------|-------|
> > > |Method|F$_\max\uparrow$|MAE$\downarrow$|F$_\max\uparrow$|MAE$\downarrow$ |F$_\max\uparrow$|MAE$\downarrow$|
> > > PVT[5]  |.900|.032 |.832 |.050 |.883 |.060 |
> > > P2T[6]  |.912|.029 |.840 |.045 |.898 |**.053**  |
> > > Ours\_RGB |**.934** |**.025** |**.866** |**.044** |**.917** |**.053** |

---

### Author Response · Authors · 2021-08-20
**A summary of the novelty and significance of the paper**

To all reviewers, ACs, and PCs:

We want to highlight the contribution of our paper again.

Our paper is novel and significant in the fields of both machine learning and computer vision.

From the machine learning (i.e., generative modeling and learning) perspective, **our model is the first top-down conditional generative model with a vision-based transformer as a decoder and an energy-based model as the informative prior distribution**. It belongs to a new member of the generative transformer family, and also a new member of the latent space EBM family, as well as a member of the conditional generative model family. As to the learning aspect, **our model is a likelihood-based deep generative model, which is neither GAN nor VAE, and is trained without relying on any assisting network**. The learning algorithm derived from the proposed model is based on MCMC inference, which is more natural, principled, and statistically rigorous.  Such a framework is not only useful for saliency prediction (this is what we target in our work) but also applicable to a vast of conditional learning scenarios. The proposed model and the learning algorithm are generic and universal.

From the computer vision (i.e., a concrete application) perspective, our model with a special network design to handle saliency prediction is a new member of the generative saliency prediction methods.  Compared with the traditional discriminative saliency prediction methods and the existing generative ones, our generative method not only shows that it is more natural and reasonable to model the saliency prediction as a conditional probability distribution but also demonstrates the state-of-the-art performances over all RGB and RGB-D benchmarks. As we know, the computer vision community has started to develop vision transformers for classification, segmentation, detection, generation, etc in the past few months. **Our paper is the first generative saliency transformer**. Thus, the model is significantly important for the computer vision community.

Our model is significantly different from all existing models in machine learning (generative models) and computer vision (saliency prediction).

**We hope all reviewers can carefully read our detailed feedbacks that have addressed your concerns and re-consider your score to give a new model a chance.**

In the meanwhile, feel free to ask more questions if you have any time. We are always happy to have a further discussion and answer more questions raised from you.

Thanks again.

Authors

---

### Decision · Program_Chairs · 2021-09-27

**Decision:**

Accept (Poster)

**Comment:**

This paper describes a novel transformer-based latent variable model for salient object detection. Reviewers were generally positive on the approach and results, and the work was deemed to be technically sound. The reviews and authors had a productive discussion which we hope will lead to some clarifications that will enhance the paper. One reviewer provided a score below the threshold for acceptance, but this reviewer appears to have had some confusion about the nature of the method and did not reengage with authors when they sought to clarify; as such, this review was down-weighted in the overall evaluation. All in all, this is an interesting method producing state of the art results, with solid theoretical grounding.